



# Temperature signal in suspended sediment export from an Alpine catchment

Anna Costa[1], Peter Molnar[1], Laura Stutenbecker[2], Maarten Bakker[3], Tiago A. Silva[4], Fritz Schlunegger[2], Stuart N. Lane[3], Jean–Luc Loizeau[4], Stéphanie Girardclos[5]

[1]Institute of Environmental Engineering, ETH Zurich, 8093 Zurich, Switzerland
[2]Institute of Geological Sciences, University of Bern, 3012 Bern, Switzerland
[3]Institute of Earth Surface Dynamics, University of Lausanne, 1015 Lausanne, Switzerland
[4]Department F.–A. Forel, University of Geneva, 1290 Versoix, Switzerland
[5]Department of Earth Sciences and Institute for Environmental Sciences, University of Geneva, 1205 Geneva, Switzerland

*Correspondence to*: Anna Costa (costa@ifu.baug.ethz.ch)

**Abstract**

Suspended sediment export from large Alpine catchments ($> 1000$ km$^2$) over decadal timescales is sensitive to a number of factors, including long–term variations in climate, the activation–deactivation of different sediment sources (proglacial areas, hillslopes, etc.), transport through the river system, and potential anthropogenic impacts on the sediment flux (e.g. through

impoundments and flow regulation). Here, we report on a marked increase in suspended sediment concentrations observed close to the outlet of the upper Rhône River Basin in the mid–1980s. This increase coincides with a statistically significant step–like increase in basin–wide mean air temperature. We explore the potential explanations of the suspended sediment rise in terms of discharge (transport capacity) change, and the activation of different sources of fine sediment (sediment supply) in the catchment by hydroclimatic forcing. Time series of precipitation and temperature–driven snowmelt, snow cover and

ice–melt simulated with a spatially distributed degree–day model, together with erosive rainfall on snow–free surfaces, are tested as possible reasons for the rise in suspended sediment concentration. We demonstrate that the abrupt change in air temperature reduced snow cover and the contribution of snowmelt, and enhanced ice–melt. The results of statistical tests showed that the onset of increased ice–melt was likely to play a dominant role in the suspended sediment concentration rise in the mid–1980s. Temperature–driven enhanced melting of glaciers, which cover about 10% of the catchment surface, can

increase suspended sediment yields through increased runoff from sediment–rich proglacial areas, increased contribution of sediment–rich meltwater, and increased sediment supply in proglacial areas due to glacier recession. The reduced extent and duration of snow cover in the catchment may also have partly contributed to the rise in suspended sediment concentration through hillslope erosion by rainfall on snow free surfaces, and by reducing snow cover on the surface of the glaciers and thereby increasing meltwater production. Despite the rise in air temperature, changes in mean discharge in the mid–1980s

were statistically insignificant, and their interpretation is complicated by hydropower reservoir management and the flushing operations at intakes. Thus, the results show that to explain changes in suspended sediment transport from large Alpine catchments it is necessary to include an understanding of the multitude of sediment sources involved together with the hydroclimatic conditioning of their activation (e.g. changes in precipitation, runoff, air temperature). This is particularly relevant for quantifying climate change and hydropower impacts on streamflow and sediment budgets in high Alpine

catchments.

## 1. Introduction

Erosion processes and sediment dynamics in Alpine catchments are determined by geological, anthropogenic, and climatic factors. Geological forcing is one of the main drivers of sediment production and landscape development, through crustal thickening, deformation and isostatic uplift, and glacier inheritance (e.g. England and Molnar, 1990; Schlunegger and

Hinderer, 2001; Vernon et al., 2008). Almost continuous glacier recession in the European Alps since the late 19th Century




(Paul, 2004; 2007; Haeberli, 2007) has maintained large parts of the landscape in early stages of the paraglacial phase where unstable or metastable sediment sources (Ballantyne, 2002; Hornung et al., 2010) can maintain high sediment supply rates. Glacier inheritance influences sediment production and transport as demonstrated by a strong spatial association between sediment yield and past and current glacial cover (Hinderer et al., 2013; Delunel et al., 2014). Anthropogenic impacts on

sediment yields are more recent, and on a global scale connected to land cover change through intensified agriculture and the trapping of sediment in reservoirs (e.g. Syvitski et al., 2005). Land use changes impact mainly fine sediment production (e.g. Foster et al. 2003; Wick et al., 2003), while river channelization, flow regulation, water abstraction, and sediment extraction have caused a general reduction in sediment yield and consequently led to sediment–starved rivers world–wide (Kondolf et al., 2014). In Alpine catchments, the situation is often complicated by flow abstraction at hydropower intakes. The reduction

of sediment transporting capacity downstream of intakes and the periodic flushing of sediment stored has severe impacts on the sediment budget (e.g. Anselmetti et al., 2007) and downstream river ecology (e.g. Gabbud and Lane, 2016).

In this paper we focus on the dominant role of climate in sediment production and transfer in Alpine environments (e.g. Huggel et al., 2012; Zerathe et al., 2014; Micheletti et al., 2015; Palazon and Navas, 2016; Wood et al., 2016). In such environments it is important to understand both the role of intense rainfall which triggers erosion and mass movements on

hillslopes (landslides, debris flows, etc.), as well as air temperature and (solid, liquid) precipitation, which may drive the hydrological processes that generate runoff (e.g. Quinton and Carey, 2008) and so determine the capacity of the rivers to transport sediment. Snow cover accumulation and melt, snow avalanching, as well as the seasonal dynamics of ice–melt, also directly affect sediment supply (e.g. Moore et al., 2013).

To assess the relative contribution of climatic variables on sediment production and yield, it is useful first to identify the

sediment sources on a catchment scale. In an Alpine catchment we typically identify four main sediment sources (Fig. 1): glacial erosion, hillslope erosion, channel bed/bank erosion and mass wasting events (e.g. rockfalls, debris flows). Climatic conditions, specifically precipitation and air temperature, contribute to the activation of these four sediment sources through different processes and at different rates (Fig. 1). Erosive processes of abrasion, bed–rock fracturing and plucking at the base of glaciers provide proglacial areas with a large amount of sediment (Boulton, 1974). Due to glacial erosion, discharge from

subglacial channels has high suspended sediment concentrations (e.g. Aas and Bogen, 1988). Temperature–driven snow and ice–melt in spring and summer, as well as intense rainfall on snow–free surfaces, may lead to entrainment from proglacial areas provided they are connected to the river network (Lane et al., 2016). Catchment hillslope erosion is mainly driven by overland flow and rainfall erosivity (Wischmeier, 1959), which may be exacerbated in Alpine catchments by permanently or partially frozen ground (Quinton and Carey, 2008). Precipitation and snowmelt generate runoff that transports sediment in

rills and gullies along snow–free hillslopes. Especially intense rainfall events in summer and autumn when Alpine catchments are largely free of snow may erode large amounts of sediments. Rainfall is also responsible for triggering mass wasting events, such as debris flows and landslides, where a large mass of sediment is delivered to the channel network instantaneously (e.g. Bennett et al., 2012). Flow conditions (e.g. shear stress, stream power) then determine the sediment transporting capacity and in–stream sediment mobilization along rivers, and hence its connection or transfer to downstream

locations.

The close link between precipitation, air temperature, runoff and the activation–deactivation of sediment sources in Alpine catchments becomes critical in the context of climate change. Alpine regions represent a very sensitive environment in relation to current rapid warming. In Switzerland, substantial glacier retreat has been measured since 1850, and considerably higher ice loss rates are found after the mid–1980s (Paul, 2004; 2007; Haeberli, 2007). A reduction in snow cover duration

and mean snow depth has been observed during the last thirty years (e.g. Beniston, 1997; Laternser and Schneebeli, 2003; Scherrer et al., 2004; Marty, 2008; Scherrer et al., 2006). Although current effects of climate change are much less clear for precipitation (Brönnimann et al., 2014) than for temperature, a sharp reduction in the number of snowfall days has been observed at many Swiss meteorological stations (Serquet et al. 2011).





The premise behind this paper is that to explain impacts of climate on Alpine catchment suspended sediment yield, it is necessary to consider at least (a) the sediment sources involved; and (b) the hydroclimatic conditioning of their activation (e.g. precipitation, runoff and air temperature). The traditional rating–curve analysis, which relates suspended sediment concentration only to discharge (e.g. Campbell and Bauder, 1940; Walling, 1974; 1977; Asselman, 1999; 2000; Lenzi and

Marchi, 2000; Horowitz, 2003; Mao and Carrillo, 2015), is insufficient for analyzing detailed changes in sediment yield because sediment supply is not explicitly accounted for (e.g. Walling, 2005; De Vente et al., 2006; Mao and Carrillo, 2015). In this study we show the need to look at specific sediment sources and their activation when analysing changes in suspended sediment transport from catchments. A more process–based perspective in the estimation of suspended sediment concentration and fine sediment yield allows us to infer effects of changes in hydroclimate, such as increases in temperature

and/or precipitation intensity, which is not possible with traditional methods based on discharge alone.

The upper Rhône River Basin draining into Lake Geneva in Switzerland is at the center of our investigation. The basin has experienced a sudden rise in air temperature that coincided with a rise in suspended sediment concentrations in the mid–1980s. Our main objective is to explore the signal of a warmer climate in the suspended sediment dynamics of this regulated and human–impacted Alpine catchment. In this work, we refer to fine sediment as the sediment transported in suspension.

We investigate the potential causes of the observed increase in suspended sediment concentration by focusing on two factors: transport capacity and sediment supply. For transport capacity we analyze daily streamflow at the catchment outlet, while for sediment supply we conceptualize the upper Rhône Basin as a series of spatially distributed sediment sources (Fig. 1) that are activated or deactivated throughout the hydrological year by hydroclimatic forcing. Hydroclimatic forcing is given by daily basin–wide observed precipitation, and simulated snow cover, snowmelt and ice–melt. Finally, we conduct

statistical tests to explore the significance of changes in the elaborated hydroclimatic variables in concordance with the observed changes in suspended sediment concentration.

Our aims are: (a) to estimate daily hydroclimatic variables over the Rhône Basin for the last 40 years, which explicitly address the activation of the four fine sediment sources in Fig. 1; (b) to provide statistical evidence for possible reasons for the rise in suspended sediment concentrations in the mid–1980s; and (c) to discuss the results in the context of future climate

change and hydropower operation impacts on sediment budgets. Although our results apply specifically to the Rhône Basin, the methodology and approach is generally valid for any Alpine catchment with similar sediment sources and hydroclimatic forcing.



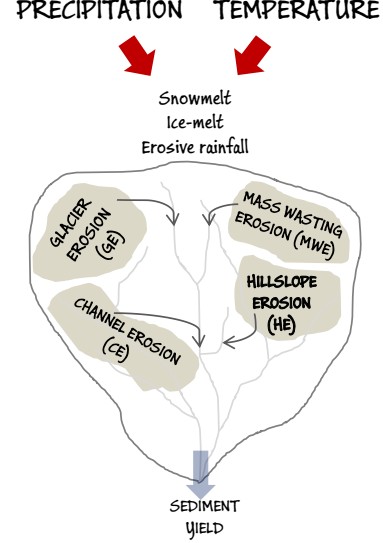

**Figure 1: Scheme representing the four main sediment sources of Alpine catchments: glacial erosion (GE), mass wasting erosion (MWE), channel erosion (CE) and hillslope erosion (HE). Precipitation and temperature drive snowmelt, ice–melt and erosive rainfall through precipitation and snow cover dynamics. These hydroclimatic variables influence the activation-deactivation of sediment sources.**

## 2. Study Site Description

The upper Rhône Basin is located in the southwestern part of Switzerland, in the Central Swiss Alps (Fig. 2). It has a total surface area of $5\,338$ km$^2$, and an altitudinal range of 372 to 4,634 m a.s.l.. About 10% of the surface is covered by glaciers, which are mostly located in the eastern and southeastern part of the catchment (Fig. 2) (Stutenbecker et al., 2016). The Rhône River originates at the Rhône Glacier and flows for about 160 km through the Rhône valley before entering Lake Geneva, a few kilometers downstream of the gauging station at la Porte-du–Scex. Basin–wide mean annual precipitation is about $1\,400$ mm yr$^{-1}$ and shows strong spatial variability driven mostly by orography and the orientation of the main valley. The hydrological regime of the catchment, typical of Alpine environments, is strongly influenced by snow and ice–melt with highest discharge in summer and lowest in winter. Mean annual discharge is 180 m$^3$ s$^{-1}$, which corresponds to about $1\,060$ mm yr$^{-1}$ and a runoff coefficient of 75%.

The catchment has been strongly affected by anthropogenic impacts during the last century. The main course of the Rhône River has been extensively channelized for the purposes of flood protection: levees were constructed and the channel was narrowed and deepened in the periods 1863–1894 and 1930–1960 (First and Second Rhône Corrections). Due to the residual flood risk that affects the main valley, a third project was started in 2009 with the main objectives to increase channel conveyance capacity and river ecological rehabilitation (Oliver et al., 2009). In addition, significant gravel mining operations are carried out along the main channel and many tributaries. Since the 1960s, several large hydropower dams have been built along the main tributaries of the Rhône River. The total storage capacity of these reservoirs corresponds to about 20% of the mean annual streamflow (Loizeau and Dominik, 2000). Flow impoundment, water abstraction and diversion through complex networks of intakes, tunnels and pumping stations, have significantly impacted the flow and sediment regime of the catchment. Flow regulation due to hydropower production has resulted in a considerable decrease of discharge in summer, increase in winter, and has partly affected the flood regime (Loizeau and Dominik, 2000). The construction of dams and start of hydropower operation has coincided with a drop in the suspended sediment load of the main Rhône River measured at la Porte-du-Scex (Loizeau et al., 1997; Loizeau and Dominik, 2000). However, the impacts of dams and reservoirs on





suspended sediment fluxes occurred in parallel with progressive climate warming, so the partitioning of human and climate impacts remains an open question. This sensitivity to climate in the post–dam period is the focus of our investigation.

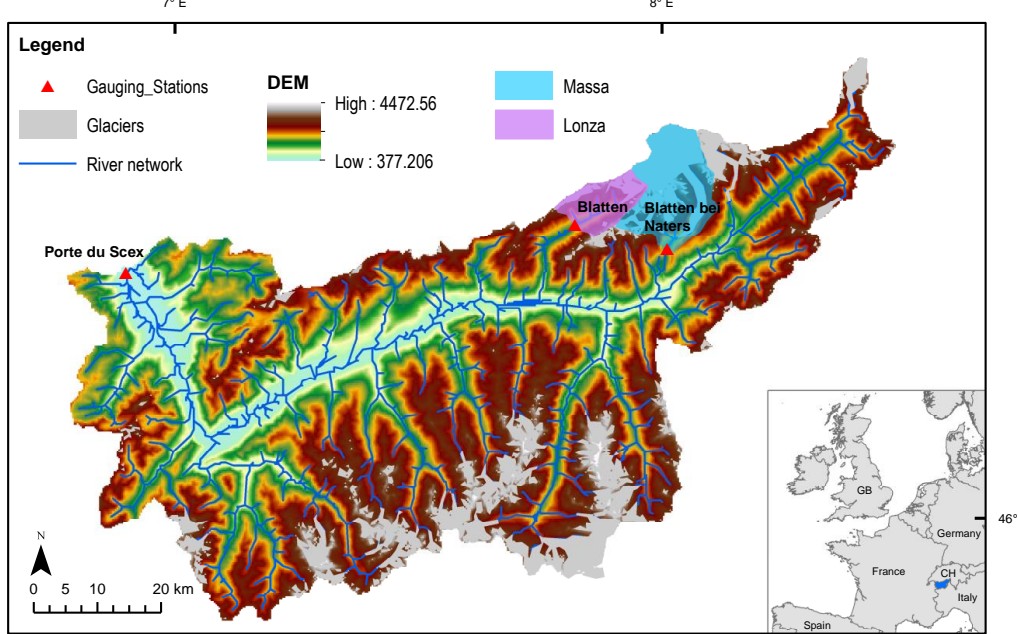

**Figure 2: Map of the upper Rhône Basin with topography, glaciated areas and river network. Inset shows the position of the upper Rhône Basin in Europe (blue). Locations of gauging stations used in this analysis are shown as triangles. Massa and Lonza sub–basins, involved in calibration and validation of the ice melting component are highlighted.**

## 3. Methods

Our objective is to explore the potential effect of climate on suspended sediment dynamics of the upper Rhône Basin in the period 1975–2015. To this end we use observed and simulated hydroclimatic and sediment yield variables: temperature $T$, precipitation $P$, discharge $Q$, suspended sediment concentration SSC, snow cover fraction SCF, snowmelt SM, ice–melt IM, and effective rainfall ER. This selection of variables is based on the consideration that three main hydroclimatic factors (ice–melt, snowmelt and rainfall) are responsible for activating sediment sources (Fig 1, Table 1) with the following processes in mind: (a) ice–melt runoff evacuates accumulated fine sediment, the product of glacial erosion (e.g. Swift et al., 2005); (b) snow cover dynamics impact ice–melt efficiency through albedo and may result in more rapid erosion and sediment production through an increased glacier basal velocity (e.g. Herman et al., 2015); (c) snowmelt runoff from snow–covered catchment areas may generate downstream hillslope erosion and channel erosion (Lenzi et al., 2003); and (d) rainfall on snow–free surfaces commonly leads to hillslope erosion, mass wasting and, due to high discharge, also channel erosion (Meusburger and Alewell, 2014).

All variables are analyzed as mean monthly and annual values for the study period averaged over the basin area and originate from observations or interpolation of observations ($T$, $P$, $Q$, SSC), from simulations by a snow and ice–melt model (SCF, SM, IM), or a combination thereof (ER). Basin–wide SM and IM are computed from simulations with a spatially distributed snow and ice–melt model based on the temperature–index method on a grid scale, while SCF is an integrated snow cover fraction at the basin scale. The snowmelt model is described in Sect. 3.1, the ice–melt model in Sect. 3.2, and their calibration in Sect. 3.3. To quantify changes in the hydroclimatic and sediment yield variables, we apply standard statistical tests for change detection described in Sect. 3.4. The data used to drive the snow and ice–melt are spatially distributed



gridded climatic datasets of mean, minimum, maximum daily air temperature and total daily precipitation on a 250×250 m grid resolution at the daily timescale over the period 1975–2015. The data are described in more detail in Sect. 4 and all variables and their use are summarized in Table 1.

5 **Table 1: Hydroclimatic variables originating from measurements or interpolation of measurements (T, P, Q, SSC), from simulations of the snow and ice–melt model (SCF, SM, IM), or a combination thereof (ER). Sediment sources affected by hydroclimatic variables are glacier erosion (GE), mass wasting erosion (MWE), hillslope erosion (HE), channel erosion (CE) as depicted in Fig. 1.**

| Variable | Data Source | Resolution | Use/Sediment Sources |
|---|---|---|---|
| T | Daily mean, minimum, maximum air temperature [° C] on a 2x2 km grid (MeteoSwiss) | daily, basin–average 1961–2015, 1975–2015 | Snowmelt model Ice–melt model |
| P | Daily mean precipitation [mm day$^{-1}$] on a 2x2 km grid (MeteoSwiss) | daily, basin–average 1961–2015 | Snowmelt model Triggering of MWE, HE |
| Q | Daily mean discharge [m$^3$ s$^{-1}$] at three stations (Porte du Scex, Blatten, Blatten bei Naters) (FOEN) | daily, 1905–2015, 1956–2015, 1930–2015 | Calibration of ice–melt model Triggering of CE |
| SSC | Suspended sediment concentration [mg l$^{-1}$] at Porte du Scex (FOEN) | 2 times per week 1964–2012 | Sediment yield |
| SCF | Snow cover fraction [0–1] simulated by the Snowmelt Model and calibrated with MODIS satellite data for the period 2000–2009 | daily, basin–average 1975–2012 | Triggering of HE, GE |
| SM | Snowmelt rate [mm day$^{-1}$] simulated by the Snowmelt model | Daily, basin–average 1975–2012 | Triggering of HE, CE |
| IM | Ice–melt rate [mm day$^{-1}$] simulated by the Ice–melt model and calibrated at Blatten and Blatten bei Naters | Daily, basin–average 1975–2012 | Triggering of GE |
| ER | Mean annual effective rainfall [mm day$^{-1}$] (rainfall on snow–free pixels) | Daily, basin–average 1975–2012 | Triggering of MWE, HE, CE |

### 3.1 Snowmelt Model

We use a snowmelt model to predict SM and SCF over the entire basin, because snow station measurements are sparsely and
15 irregularly distributed and a consistency between precipitation and air temperature as climatic driving forces and snowmelt and snow cover as response variables are needed. The spatially distributed temperature index method (degree–day model) was used due to its simplicity, low data requirements, and demonstrated success at daily temporal scales over large basins (e.g. Hock, 2003; Boscarello et al., 2014). The degree–day approach also matches the coarse spatial (250×250 m) and temporal (daily) resolution of our analysis and the integration at the basin scale. Models based on energy balance, or
20 enhancements of the degree–day approach, represent physical processes better and should be used when higher spatial and temporal resolution and accuracy is needed (e.g. Pellicciotti et al., 2005).

The Snowmelt Model includes snow accumulation and melting. At the grid scale, precipitation $P$ [mm day$^{-1}$] is first partitioned into solid and liquid form based on daily minimum $T_{min}$ [°C], maximum air temperature $T_{max}$ [°C] and a rain–snow threshold temperature $T_{RS}$ [°C]. If minimum air temperature $T_{min}$ is above the threshold temperature $T_{RS}$, all





precipitation falls as rainfall $R$; if the maximum air temperature $T_{max}$ is below the threshold temperature $T_{RS}$, all precipitation falls as snow $S$; otherwise precipitation is a mixture of liquid and solid form, partitioned proportionally to the temperature differences:

$$\begin{cases} R = c_p P \\ S = (1 - c_p)P \end{cases},$$
(1)

where

$$\begin{cases} c_p = 1 & T_{min} > T_{RS} \\ c_p = 0 & T_{max} \leq T_{RS} \\ c_p = \frac{T_{max} - T_{RS}}{T_{max} - T_{min}} & T_{min} \leq T_{RS} < T_{max} \end{cases},$$
(2)

The daily snowmelt rate $M_{snow}$ [mm day$^{-1}$] is estimated from a linear relation with air temperature:

$$\begin{cases} M_{snow} = k_{snow}(T_{mean} - T_{SM}) & T_{mean} > T_{SM} \\ M_{snow} = 0 & T_{mean} \leq T_{SM} \end{cases},$$
(3)

where $T_{mean}$ [°C] is the mean daily air temperature, $T_{SM}$ [°C] is a threshold temperature for the onset of melt, and $k_{snow}$ is a

melt factor [mm day$^{-1}$ °C$^{-1}$]. Snow depth (SD), in mm snow water equivalent, is then simulated in time from a balance between accumulation and melting at the grid scale $i$:

$$SD_i(t) = SD_i(t-1) + S_i(t) - M_{snow_i}(t).$$
(4)

The snow cover fraction SCF for a chosen area containing $i = 1, \ldots, N$ grids, is:

$$SCF(t) = \frac{1}{N}\sum_{i=1}^{N} H[SD_i(t)],$$
(5)

where $H$ is a unit step function: $H = 0$ when SD $= 0$ and $H = 1$ when SD $> 0$. The area of integration N can be the entire catchment, sub–basins, elevation bands, etc. For the entire catchment, we estimate mean daily snowmelt SM [mm day$^{-1}$] as the arithmetic average over all grid melt rates:

$$SM(t) = \frac{1}{N}\sum_{i=1}^{N} M_{snow_i}(t).$$
(6)

The threshold temperatures for defining the precipitation type $T_{RS}$ and the onset of melt $T_{SM}$ depend on many factors such as

atmospheric boundary layer conditions, temperature, humidity, cloud types, among others. Different parametrizations and temperature values are available in the literature (Wen et al., 2013). Depending on region, altitude and modelling approach, rain–snow temperature thresholds show a wide range of variability from −5 °C (Collins et al., 2004) to more than 6 °C (Auer, 1974). For the upper Rhône Basin we assume a constant rain–snow temperature threshold $T_{RS} = 1$°C, resulting from a calibration and validation of the physically–based fully distributed hydrological model Topkapi–ETH in the catchment

(Fatichi et al., 2015). To reduce degrees of freedom, the threshold temperature for the onset of melt $T_{SM}$ is set equal to 0°C and the calibration of the snowmelt model consists of estimating the melt factor $k_{snow}$ with methods described in Sect. 3.3.

### 3.2 Ice–melt Model

Similar to snowmelt, ice–melt is also simulated with a temperature index (degree–day) model on grid cells that are identified

as glacier–covered from the GLIMS dataset. The ice–melt rate $M_{ice}$ [mm day$^{-1}$] on glacier surfaces that are snow–free is estimated as:

$$\begin{cases} M_{ice} = k_{ice}(T_{mean} - T_{IM}) & T_{mean} > T_{IM} \\ M_{ice} = 0 & T_{mean} \leq T_{IM} \end{cases},$$
(7)

where $T_{mean}$ [°C] is mean daily air temperature, $T_{IM}$ [°C] is a threshold temperature for the onset of ice–melt, and $k_{ice}$ [mm day$^{-1}$ °C$^{-1}$] is the ice–melt factor. For the entire catchment, we estimate mean daily ice–melt IM [mm day$^{-1}$] as the arithmetic

average over all ice–covered grids:

$$IM(t) = \frac{1}{N}\sum_{i=1}^{N} M_{ice_i}(t).$$
(8)

The threshold temperature for glacier melting $T_{IM}$ is set equal to 0 °C. Ice–melt occurs only if the glacier cell is snow–free, therefore grid snow cover dynamics on glacier–covered pixels simulated by the snowmelt model in Sect. 3.1, are essential





for estimating ice–melt. The calibration of the ice–melt model consists of estimating the melt factor $k_{ice}$ with data from two highly glaciated sub–basins in the Rhône (Massa and Lonza) shown in Fig. 2, with methods described in Sect. 3.3.

It should be noted that we do not consider glacier dynamics in this study, i.e. changes in ice thickness due to accumulation and melting, as well as glacier movement. Comparison between glacier length reduction in the study period and the effective

model (grid) resolution in our analysis shows that the length reduction of Alpine glaciers reported in the literature for the last fifty years is smaller than the resolution of climatic input datasets (Hoelzle, 2003, Oerlemans, 2005). Therefore, potential overestimation of ice–melt coming from cells where the glacier melted away is considered to be negligible. Clearly, this would not be the case if we were looking at future climate change projections and at glaciers that are rapidly disappearing. Under climate change, even the largest glacier in the basin, the Aletsch Glacier, is expected to shrink at a rate where its ice–

melt contribution would start decreasing before 2050 (FOEN, 2012; Brönnimann et al., 2014).

### 3.3 Calibration and Validation

Calibration and validation of the snow and ice–melt model parameters are performed in sequence, since the snow–covered surface is required for ice–melt estimation on glaciers. The snowmelt factor $k_{snow}$ is calibrated based on comparisons with maps of snow cover derived from satellite images (MODIS). Snow cover observations are split into two periods: 1 October

2000 – 30 September 2005 for calibration and 1 October 2005 – 31 December 2008 for validation. MODIS maps of snow cover are filtered to reduce the impacts of clouds on snow cover fraction estimations. The resulting number of calibration and validation days is equal to 217 and 143 respectively. Snow cover maps at 500×500 m resolution are distributed by proximal interpolation to the snowmelt model's 250×250 m computational grid. Snow depth maps simulated with Eq. (4)

and transformed into a simulated snow cover fraction SCF$^{sim}$ in Eq. (5) are compared with the MODIS maps SCF$^{obs}$.

The quantification of the errors and objective function for calibration is based on a combination of mean absolute error and true skill statistic. The mean absolute error MAE is estimated as:

$$\text{MAE} = \frac{1}{n} \sum_{j=1}^{n} \left| \text{SCF}_j^{obs} - \text{SCF}_j^{sim} \right| , \tag{9}$$

where n is the number of MODIS image maps, and it captures the overall ability of the model to reproduce the snow cover

fraction accurately. The true skill statistic TSS is a spatial statistic that measures the grid–to–grid performance of the model in capturing snow–no snow presence. It is computed as the sum of sensitivity SE (correct snow predictions) and specificity SP (correct no–snow predictions) from contingency tables (e.g. Wilks, 1995; Mason and Graham, 1999; Corbari, 2009):

$$\text{TSS} = \frac{1}{n} \sum_{j=1}^{n} \text{TSS}_j = \frac{1}{n} \sum_{j=1}^{n} \text{SE}_j + \text{SP}_j - 1 \tag{10}$$

in each image j and averaged over the n MODIS maps in the simulation period. Because TSS includes both sensitivity and

specificity, this measure captures both predictions of snow–covered and snow–free areas, and it measures the accuracy of the model against the level of agreement that would be expected by chance. It takes on values between 0 and 1, where 1 indicates perfect performance. TSS is widely applied in assessing spatial model performance (e.g. Begueria, 2006; Allouche et al., 2006). Finally, both goodness–of–fit measures are combined into an objective function OF, giving slightly more weight to MAE and evaluated over k = 5 different elevation bands in order to better capture the topographic gradients in

snowmelt distributions in the Rhône Basin:

$$\text{OF} = \sum_{k=1}^{5} \text{OF}_k = \sum_{k=1}^{5} -0.6 \, \text{MAE}_k + 0.4 \, \text{TSS}_k . \tag{11}$$

This objective function is maximized in calibration. The rationale of using both MAE and TSS in evaluating performance is to give weight to both basin–integrated snow cover as well as to grid–based predictions. Indeed, the same value of snow cover fraction can result in two different spatial arrangements of snow–covered pixels, and a correct spatial distribution of

snow–covered and snow–free areas is relevant for this analysis insofar as it affects the activation and deactivation of specific sediment sources. The actual weights assigned to MAE and TSS terms in Eq. (11) were the outcome of sensitivity tests with





the model. After calibration we also estimated the Nash–Sutcliffe efficiency NS (Nash and Sutcliffe, 1970) and the mean square error MSE to illustrate the performance of the model:

$$\text{NS} = 1 - \frac{\sum_{j=1}^{n}(\text{SCF}_j^{\text{obs}} - \text{SCF}_j^{\text{sim}})^2}{\sum_{j=1}^{n}(\text{SCF}_j^{\text{obs}} - \overline{\text{SCF}})^2} \ , \tag{12}$$

$$\text{MSE} = \frac{1}{n}\sum_{j=1}^{n}(\text{SCF}_j^{\text{obs}} - \text{SCF}_j^{\text{sim}})^2 \ , \tag{13}$$

where $\overline{\text{SCF}}$ is the average observed snow cover fraction during the calibration–validation period.

The ice–melt factor $k_{\text{ice}}$ is calibrated on the sub–basin of the river Massa located in the north–east part of the upper Rhône Basin (Fig. 2). The Massa is a medium–sized basin (195 km$^2$) with a mean elevation of 2945 m a.s.l. More than 60% of the surface is glaciated, and the remaining surface is classified mostly as rock and firn (Boscarello et al., 2014). The basin includes the Aletsch Glacier, which is the largest glacier in the European Alps with a length of around 23.2 km and a surface

area of approximately 86 km$^2$ (Haeberli and Holzhauer, 2003). Daily discharge measurements are available at Blatten bei Naters. The gauging station is located upstream of the Gebidem dam, therefore discharge is not influenced by reservoir regulation and represents undisturbed natural flow. The calibration of $k_{\text{ice}}$ is based on daily discharge measurements, focusing only on months when the ice–melt contribution is not negligible (June–October). Calibration is performed on the period between 1 January 1975 and 31 December 2005, while validation covers the remaining ten years of available data, i.e.

the time span between 1 January 2006 and 31 December 2015. The model is then validated on a second sub–basin with the same procedures and goodness of fit measures. The Lonza is a relatively small basin located to the west of the Massa (Fig. 2) with an average elevation of 2630 m a.s.l. It has a total drainage area of 77.8 km$^2$ and its surface consists of 36% of glacier cover. Daily discharge measurements at the gauging station Blatten are used for the validation.

The optimal value of $k_{\text{ice}}$ is found by minimizing the mass balance error $\text{MBE}_s$ computed for the period June–October:

$$\text{MBE}_s = 100 \ \frac{\sum_{i=1}^{ny}(V_i^{\text{obs}} - V_i^{\text{sim}})}{\sum_{i=1}^{ny} V_i^{\text{obs}}} \ , \tag{14}$$

where ny is the number of calibration years, $V_i^{\text{obs}}$ and $V_i^{\text{sim}}$ [mm year$^{-1}$] are the observed and simulated discharge volumes per unit area reaching the outlet of the catchment during the period June–October of each calibration year i:

$$V^{\text{obs}} = \sum_{j=1}^{nd} Q_j^{\text{obs}} \ , \tag{15}$$

$$V^{\text{sim}} = \sum_{j=1}^{nd} Q_j^{\text{sim}} = \sum_{j=1}^{nd}(R_j + SM_j + IM_j) \ . \tag{16}$$

Here, nd is the number of observation days from June to October, $Q_j^{\text{obs}}$ [mm day$^{-1}$] is the daily discharge per unit area observed at Blatten Bei Naters (Blatten), $R_j$, $SM_j$, $IM_j$ are respectively the total daily rainfall, snowmelt and ice–melt aggregated over the Massa (Lonza) basin. Rainfall ($R$) and snowmelt ($SM$) are simulated with the snow accumulation and melt model in Sect. 3.1, while ice–melt ($IM$) is simulated with the ice–melt model in Sect. 3.2.

**3.4 Testing for Change**

The non–parametric Pettitt test (Pettitt, 1979) is used for the detection of the time of change (year–of–change) in the air temperature data. The other variables are then tested for changes in mean (and variance) by splitting the time series into two periods before and after the identified year–of–change, and the equality of the means (and variances) is tested with the two–sample two–sided $t$–test. The null hypothesis of no change is tested at the 5% significance level. The $t$–test is a parametric

test commonly used in hydrology and atmospheric science to assess the validity of the null hypothesis of two samples having equal means and unknown unequal variances. We apply the $t$–test to all hydroclimatic variables averaged at the annual and monthly timescales with the same year–of–change to determine which hydroclimatic variables, and therefore the activation or deactivation of which sediment sources, are possibly responsible for the observed changes in suspended sediment concentration.





## 4. Data Description

### 4.1 Precipitation and Air Temperature

For precipitation and air temperature we used spatially distributed datasets provided by the Swiss Federal Office of Meteorology and Climatology (MeteoSwiss). Total daily precipitation, mean, minimum and maximum daily air temperature

are available on a 2×2 km resolution grid for Switzerland. RhiresD v1.0 provides spatially distributed total daily precipitation as rainfall and snowfall water equivalent since 1961 (Meteoswiss, 2013). Temperature datasets TabsD v1.2, TminD v1.2, TmaxD v1.2, provide mean, minimum and maximum air temperature 2 m above ground level since 1971 (Meteoswiss, 2013). All four datasets are developed by spatial interpolation of quality–checked data collected at MeteoSwiss meteorological stations. For temperature, the spatial interpolation is derived with a deterministic method based on elevation,

developed by Frei (2014). The scheme consists of the superposition of a background temperature pattern representing the vertical gradient of temperature, and a residual temperature pattern representing deviations from the background profile at scales finer than the spacing of the stations. Specific features, such as the use of a non–linear parametric profile for temperature–elevation dependence that can reproduce thermal inversion, and the introduction of a non–Euclidean distance for the weighting distance scheme, address interpolation issues typical for mountainous regions (Frei, 2014). The

precipitation dataset is derived as a combination of a climatological mean field and anomaly field obtained by interpolating precipitation anomalies registered at measurement stations relative to mean climatology at the daily timescale (Schwarb, 2000; Frei, 2006). We apply the statistical analysis of change to basin–averaged values of precipitation and temperature and not to individual grid point values, which are potentially affected by interpolation errors. We also verified the potential effect of non–homogeneities due to the varying number of stations in time by comparison with an experimental dataset

developed by MeteoSwiss specifically for this research. This dataset is based on a constant number of stations (294 for precipitation and 48 for temperature) for the period 1971–2013.

### 4.2 Discharge and Suspended Sediment Concentration

The Swiss Federal Office for the Environment (BAFU) manages 15 gauging stations within the upper Rhône Basin. Daily discharge data, measured at three gauging stations, are used in this analysis: la Porte-du-Scex at the outlet of the catchment,

and Blatten Bei Naters and Blatten, which are located at the outlet of the Massa and the Lonza sub–basins respectively (Fig. 2).

Suspended sediment concentration data are available from October 1964 at la Porte-du-Scex, 5 km upstream of the inflow of the Rhône River into Lake Geneva. Integrated point samples are taken regularly, twice per week, at this station. Point measurements are combined with on–site cross–sectional profiles of suspended sediment concentration regularly updated to

ensure the representativeness of measurements. River cross–sectional profiles are updated twice per year to account for changes over time (Grasso et al., 2012). In this work, with the term fine sediment we refer to sediment transported in suspension. Previous analysis on the grain size distribution of suspended sediments at the outlet of the upper Rhône River report about a bimodal distribution, with mode diameters equal to 13.7 μm (silt) for the finer fraction and 39.6 μm (silt) for the coarser grains (Santiago et al., 1992). Grain sizes cover a wide range of values, including clay (16.9%), silt (64.7%) and

sand (18.4%). The mean suspended sediment size is reported to be equal to 17.7 μm (silt), and the largest grains transported in suspensions, most likely during summer high flow conditions, are in the range of coarse sand (> 500 μm) (Santiago et al., 1992).

### 4.3 Snow Cover and Glacier Data

We used snow cover maps derived from satellite imagery for the upper Rhône Basin over the period 2000–2008 processed in

previous research (Fatichi et al., 2015). The 8–day snow cover product MOD10A2 retrieved from the Moderate Resolution





Imaging Spectroradiometer (MODIS) (Dedieu et al., 2010) is used for the calibration and validation of the snowmelt model. MOD10A2 is provided at a 500×500 m spatial resolution, where cells are classified as snow–covered, snow–free, inland water or cloud–covered. In order to reduce the impacts of clouds in estimating snow cover fraction, maps with cloud cover greater than 30% are excluded from the dataset, resulting in a total number of usable images equal to 360, i.e. on the average

40 days per year.

The surface covered by glaciers is assigned based on the GLIMS Glacier Database (Fig. 2). Ice–covered cells identified based on the GLIMS data of 1991 give more than 10% of the upper Rhône Basin as covered by ice with a total glacier surface of almost 620 km$^2$. Temporal dynamics of glacier coverage are not accounted for — ice accumulation, glacier retreat and ice movement are disregarded. The reduction of Alpine glaciers for the period 1950–2000 was estimated to be within the

range 500–1000 m (Hoelzle, 2003; Oerlemans, 2005), while our effective climate grid resolution is 2×2 km, i.e. the retreat is considerably lower than the grid resolution of climatic inputs. The consideration of ice dynamics would therefore add a degree of complexity that our spatial resolution cannot take advantage of.

### 4.4 Digital Terrain Model

We used a digital terrain model (DTM) with 250×250 m resolution (85409 cells in total, Fig. 2), obtained by resampling a

finer model (25×25 m) provided from SwissTopo by the ETH geodata portal (GeoVITe). The DTM is used for height information used in snowmelt modelling and as a mask for extracting climatic inputs.

### 5. Results

### 5.1 Calibration of Snowmelt and Ice–melt Models

The calibrated snowmelt factor for the Rhône Basin, following the procedure described in Sect. 3.3, is $k_{snow} = 3.6$ mm day$^-$

$^1$ °C$^{-1}$ (Fig. 3a). This value is in agreement with previous studies carried out in this region (see discussion). The calibrated snowmelt model reproduces well the seasonal fluctuations of snow cover fraction (SCF) in the basin, with Nash–Sutcliffe efficiencies (NS) close to 0.90 and low mean square errors (MSE). The model maintains a good performance also in the validation period showing very slight reduction in the goodness of fit measures (Table 2). The temporal variability of SCF is also well simulated at the basin scale. Although the comparison between observed and simulated SCF is affected by the

discontinuous nature of the MODIS data (8–day resolution), Fig. 4 shows that the model with a single constant $k_{snow}$ reproduces the snow cover dynamics reasonably well for all of the studied elevation bands. At lower elevations, the model tends to slightly underestimate SCF in autumn and overestimate it in winter. This is likely related to errors in partitioning precipitation into solid and liquid form. The model performs better at higher elevation bands, even at the very highest elevations with permanent snow cover (Fig. 4 bottom). Good snow cover simulation results at the highest elevation

bands, where most of the glaciers are located, are a prerequisite for successful ice–melt estimation.

One of the main problems of degree–day models is related to their poor performance in reproducing the spatial distribution of snow accumulation and melt in complex topography. The temperature–index approach does not take into account features that affect melting, such as topographic slope, aspect, surface roughness and albedo (Pellicciotti et al., 2005). In our case, the spatial distribution of snow cover is satisfactory, with average values of sensitivity and specificity greater than 0.7 (Table 2).

Goodness of fit measures indicate that, on average, more than 70% of snow–covered and snow–free pixels are correctly identified. However, sensitivity and specificity are characterized by a strong seasonal signal. In summer, when a large part of the basin is snow–free, it is much easier for the model to capture snow–free pixels correctly than snow–covered pixels. In winter, when the basin is largely snow–covered, the situation is reversed. The true skill score, which combines both metrics, results in values around 0.5 (Table 2). However, snow cover duration maps averaged over the period 2000–2008 for MODIS




observations and simulations show a good spatial coherence (Fig. 5). As a result, we are confident that the snowmelt model represents the spatial and temporal dynamics of snow cover in the Rhône Basin satisfactorily.

The calibrated ice–melt factor, following the procedure described in Sect. 3.3, is $k_{ice} = 6.1$ mm day$^{-1}$ °C$^{-1}$ (Fig. 3b). Despite the large variability that characterizes melt factors, comparison with previous studies confirms that this value is reasonable

5    for the Alpine environment (see discussion). Calibration and validation results are summarized in Table 2. The seasonality in all three main components of the hydrological cycle which contribute to runoff in glaciated high Alpine sub–basins (IM, SM, R) is shown in Fig. 6. The fit to the observed discharge is very good with the calibrated $k_{ice}$, with mass balance errors about 7% for the Massa and 11% for the Lonza sub–basins.

10   **Table 2.** Top: calibrated value of snowmelt ksnow factor and goodness of fit measures for validation and calibration period: Nash–Sutcliffe efficiency (NS), mean square error (MSE), true skill statistic (TSS), sensitivity (SE) and specificity (SP) for the entire upper Rhône Basin. Bottom: calibrated value of ice–melt factor kice and goodness of fit measures: mass balance error computed on June–October months (MBES) and on the entire year (MBEA) for Massa and Lonza sub–basins.

|  | $k_{snow} = 3.6$ mm day$^{-1}$ °C$^{-1}$ | |
|---|---|---|
|  | Calibration | Validation |
| NS | 0.88 | 0.86 |
| MSE | 0.01 | 0.01 |
| TSS | 0.54 | 0.46 |
| SE | 0.77 | 0.76 |
| SP | 0.73 | 0.70 |
|  | $k_{ice} = 6.1$ mm day$^{-1}$ °C$^{-1}$ | |
|  | MBE$_S$ [%] | MBE$_A$ [%] |
| Calibration Massa | 6.10 | 7.22 |
| Validation Massa | 6.77 | 9.19 |
| Validation Lonza | 11.35 | 10.09 |

(a)                                          (b)

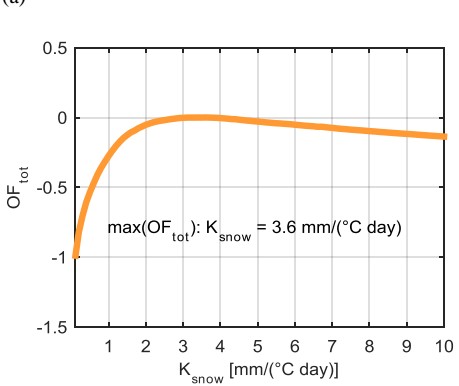
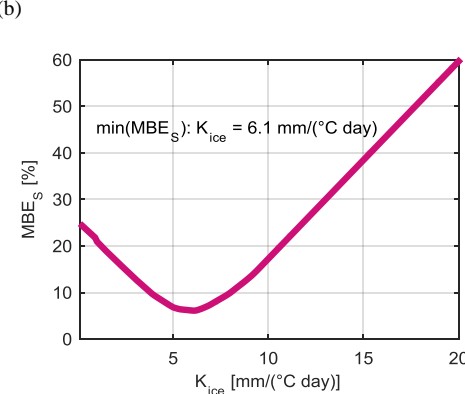

**Figure 3:** Calibration of snow and ice–melt factors: (a) global objective function (OF$_{tot}$) and $k_{snow}$ for the upper Rhône Basin; (b) mass balance error (MBE$_S$) as function of $k_{ice}$ for the Massa sub–basin. Maximum value of OF$_{tot}$ and minimum value of MBE$_S$ give $k_{snow} = 3.6$ mm day$^{-1}$ °C$^{-1}$ and $k_{ice} = 6.1$ mm day$^{-1}$ °C$^{-1}$.





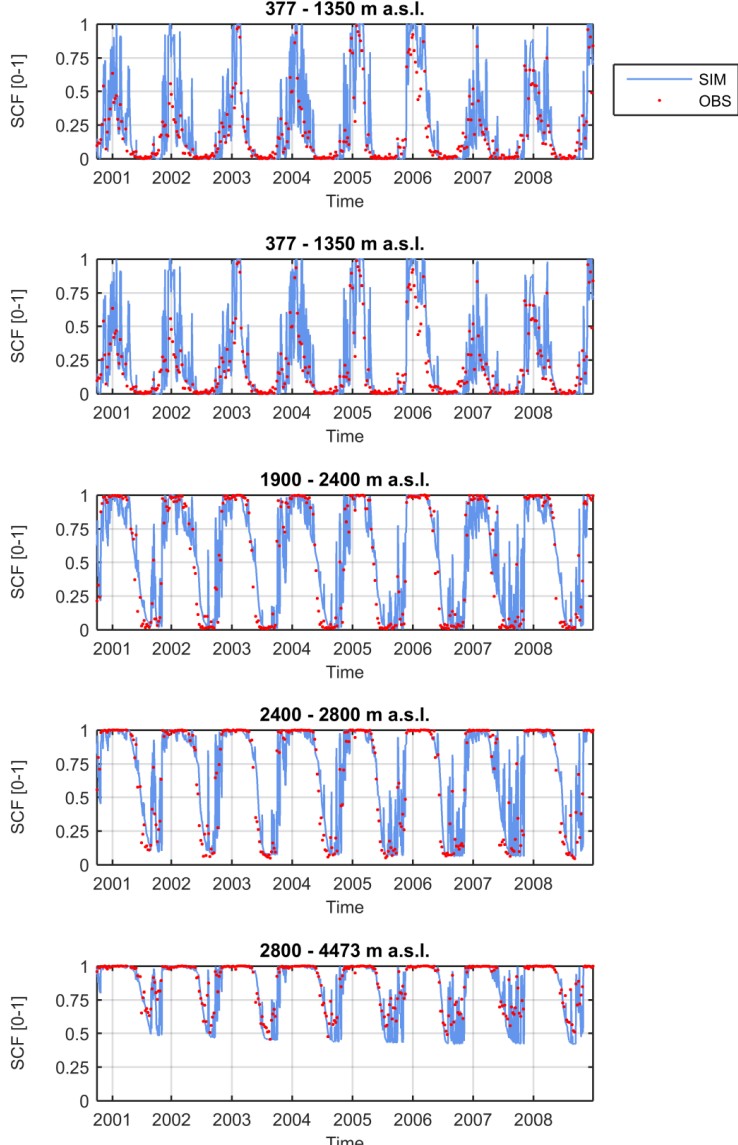

**Figure 4: Comparison between observed (red) and simulated (light blue) snow cover fraction (SCF) of the upper Rhône Basin for five different elevation bands. Simulations are computed with calibrated snowmelt factor Ksnow = 3.6 mm day–1 °C–1.**





(a)                                                              (b)

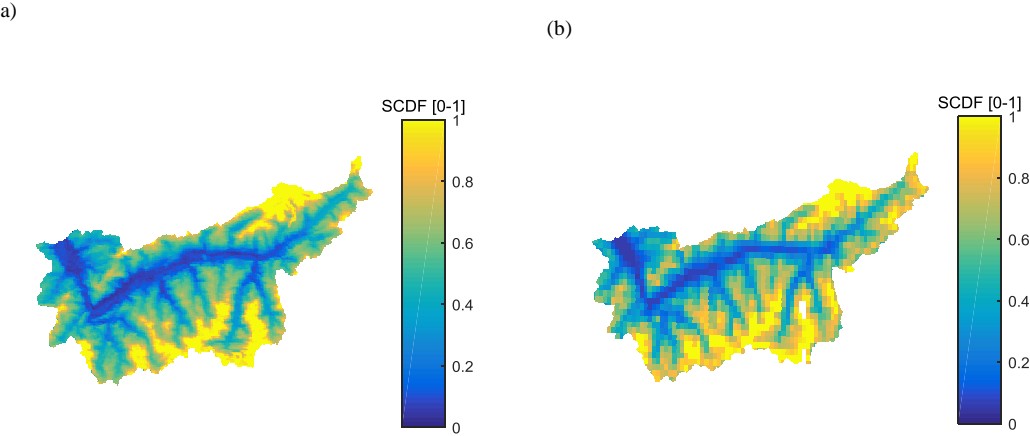

**Figure 5: Map of average snow permanence over the period 2000–2008, expressed as fraction of time in which pixels are snow–covered (SCDF [0 – 1]): (a) simulations and (b) observations (MODIS).**

(a)                                                              (b)

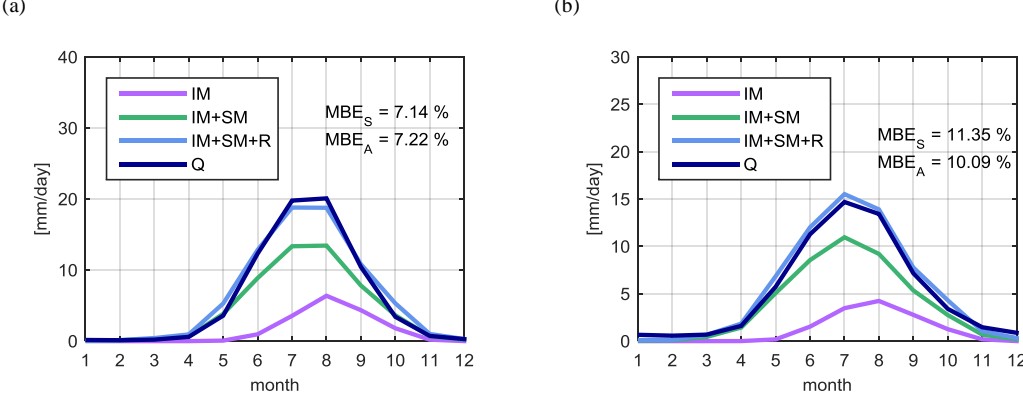

**Figure 6: Comparison of mean monthly observed (dark blue) and simulated discharge (light blue) for the period 1975–2015: (a) Massa basin and (b) Lonza basin. Simulated discharge is the sum of three components: ice–melt (IM), snowmelt (SM) and rainfall (R).**

## 5.2 Temperature, Precipitation, and SSC in the Rhône Basin

Mean annual air temperature shows a clear and statistically significant increase in 1987 (p–value < 0.01). A two–sample $t$–

test for equal means (p–value < 0.01) confirms an increase in mean daily temperature greater than 1 °C (Fig. 7a). Such dramatic jumps rather than a gradual change in air temperature have been observed globally (e.g. Jones and Moberg, 2003; Rebetez and Reinhard, 2008). Observations indicate that Switzerland has experienced two main rapid warming periods in the past, with the 1940s and 1980s being the warmest decades of the last century (Beniston et al., 1994; Beniston and Rebetez, 1996). Statistical tests on monthly means reveal that the 1987 temperature jump is mainly in spring and summer months

from March to August, while changes in the autumn and winter months are not statistically significant (Fig. 8a). For the period March-August, mean monthly temperatures have risen by about 1.2 °C on the average.




The change in air temperature around 1987 coincides with statistically significant changes in suspended sediment concentration, which has increased by about 70 mg l$^{-1}$ (Fig 7c). This change can be ascribed to statistically significant (p–value < 0.01) increases in summer (July–August) concentrations (Fig 8c). After the abrupt warming, mean annual suspended sediment concentrations are roughly 40% larger than before, with average values rising from 172 ± 6.86 mg l$^{-1}$ before 1987

up to 242 ± 14.45 mg l$^{-1}$ after 1987, where the ranges express the standard error of the mean. The simultaneous increase in temperature and suspended sediment concentration indicates that changes in climatic conditions may effectively impact sediment dynamics, especially in Alpine environments where temperature–driven processes, like snow and ice–melt, have a strong influence on the basin hydrology. Suspended sediment concentration is also characterized by a much larger inter–annual variability after 1987 than before. A statistically significant change in the variance supports the finding that processes

related to fine sediment regime of the upper Rhône Basin have been altered by changing climatic conditions, resulting in a larger concentration and variability in suspended sediment reaching the outlet of the basin.

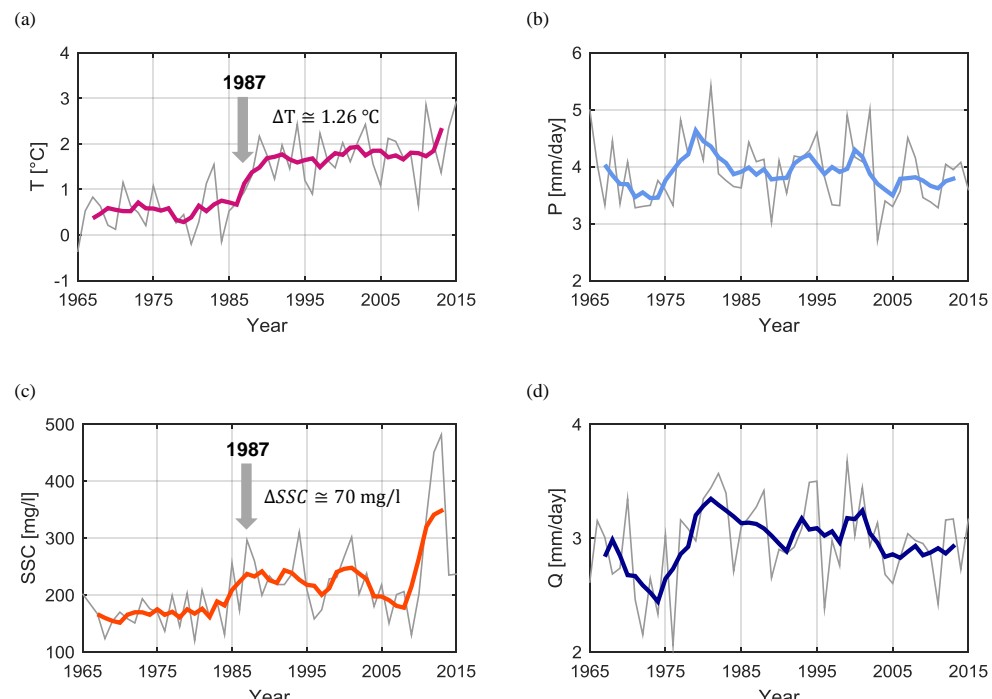

**Figure 7: Observations for the period 1965–2015: (a) average basin air temperature; (b) average basin daily precipitation; (c) mean annual suspended sediment concentration measured at the outlet of the basin; (d) daily discharge per unit area measured at**

**the outlet of the basin. Mean annual values are shown in grey and a 5–years moving average is shown with a thick line.**

While the upper Rhône Basin has undergone an abrupt warming around 1987, mean annual precipitation (Fig 7b) and monthly precipitation (Fig 8b) do not change significantly in time, and neither have mean annual discharge (Fig 7d) and monthly discharge (Fig 8d). The latter shows more variability in time and a small statistically significant increase in winter

(November-February) runoff, most likely due to increased snowmelt and possibly changes in hydropower generation. Mean annual precipitation shows rather large variability due to the complex climatology that characterizes the European Alps (Bartolini et al., 2009), and this variability is reflected in streamflow. Periodic decadal–scale oscillations appear to be present in both precipitation and runoff. These may be caused by large–scale climatic patterns, e.g. North Atlantic Oscillation (NAO), especially during winter months (Hurrell, 1995; Hurrell et al., 2003; Casty et al., 2005). Positive phases of NAO are




associated with increased moist and warm air over Western Europe and consequent enhanced winter precipitation in Scandinavia and lower precipitation in Southern Europe (Hurrell, 1995; Hurrell et al., 2003; Bartolini et al., 2009). Although the Alpine region is recognized to be in a transition region that weakens the effect of NAO on climatic conditions (Casty et al., 2005; Bartolini et al., 2009), a correlation between decadal–frequency oscillations patterns and climatic features has been

demonstrated in some Swiss locations (Beniston and Jungo, 2002).

In summary, differences in precipitation before and after 1987 are within the 95% confidence interval and are not statistically significant. Differences in discharge are also not statistically significant except in winter, when the suspended sediment concentration doesn't show changes. Therefore, it is very unlikely that the abrupt increase in suspended sediment concentration around mid-1980s in July and August is caused by changes in mean precipitation and discharge, i.e. transport

capacity.

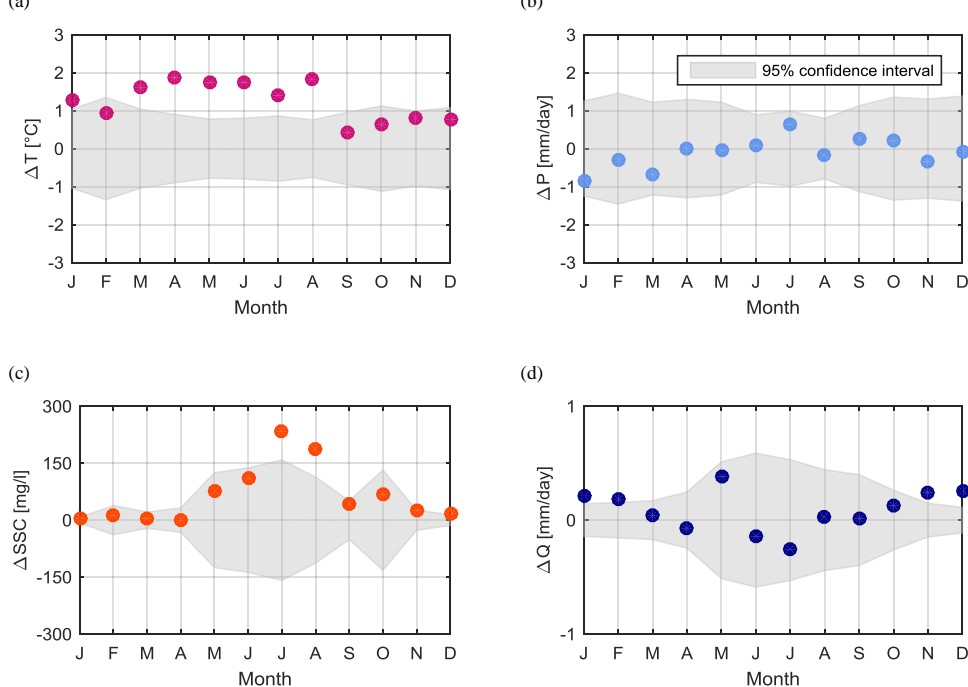

**Figure 8: Monthly differences between the period after and before the year–of–change (1965–1986 and 1987–2015) for: (a) average basin air temperature; (b) average basin daily precipitation; (c) mean suspended sediment concentration measured at the outlet of**
**the basin; (d) daily discharge per unit area measured at the outlet of the basin. Points outside the confidence interval (grey shaded area) represent statistically significant (5% significance level) changes in the monthly mean.**





### 5.3 Hydroclimatic Activation of Sediment Sources

The fact that sediment supply impacts suspended sediment concentration is evident in the sediment rating curve $SSC = aQ^b$ (Fig. 9) where large variability is evident. Given the same discharge conditions (transport capacity), actual suspended sediment concentration depends on many factors, most importantly the spatial location of sediment sources (e.g. different lithology, distance to outlet, connectivity) and the specific processes of sediment production (e.g. hillslope erosion, glacial erosion, release of subglacially stored sediment, channel bed and bank erosion, mass wasting events), all of which contribute to the variability around the sediment rating curve. Changes in fine sediment dynamics of the Rhône Basin observed during mid–1980s are more likely related to altered sediment supply conditions than to a larger transport capacity. We consider here the three main sediment production and transfer processes typical of Alpine environments: (1) the continuous effect of snowmelt runoff on hillslope and channel erosion, (2) the intermittent but potentially considerable contribution of hillslope, channel bed and bank erosion, and mass wasting triggered by rainfall events, (3) the sediment–rich flux coming from glaciated areas during the ice–melt season. To identify possible changes in sediment fluxes driven by the warming around 1987, we analyse the time series of simulated snowmelt, snow cover fraction, ice–melt and effective precipitation for annual (Fig. 10) and for monthly (Fig. 11) timescales before and after this year.

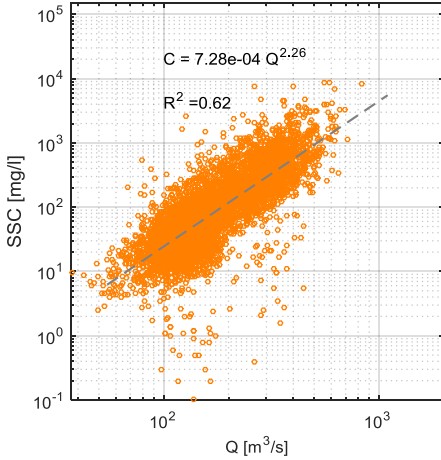

**Figure 9: Suspended sediment concentration and discharge data, measured at la Porte-du-Scex, the outlet of the upper Rhône Basin, during the period 1965–2015 together with a fitted sediment rating curve.**

First, mean annual simulated snowmelt (SM) shows a decreasing tendency during the last thirty years (Fig. 10a). The reduction in snowmelt after 1987 occurs mostly in summer and early autumn (Fig. 11a) mainly due to poor snow cover. However, except July and September, the changes in all months are within the 95% confidence interval. The increase of snowmelt in March and April is due to warmer temperatures in spring. Results are coherent with the temporal evolution of simulated snow cover fraction, which is also gradually decreasing (Fig. 10b), especially in spring and summer (Fig. 11b). Statistical analysis reveals a step–like reduction of more than 10% for mean annual values of snow cover fraction in 1987 (p–value < 0.01). Our simulations are in agreement with snow observations across Switzerland (see discussion). The reduction in snow cover duration and snow cover fraction is not relevant only for the snowmelt component, but it has also a significant impact on ice–melt by enhancing ice–melting efficiency and so subglacial suspended sediment release, and on rainfall erosion by increasing potential effects of direct rainfall on snow–free surfaces.




Second, although mean annual and monthly precipitation were shown not to change significantly in the mid–1980s, effective rainfall (ER) on snow free areas has increased, especially in early summer (Fig. 10d, Fig. 11d). The direct impact of rainfall on sediment detachment and transport depends on precipitation form, soil and land cover, as well as snow cover which protects the soil from the erosive effect of rainfall. Therefore, ER provides a metric for evaluating changes in the erosive

power of rainfall during last forty years. The effective rainfall increases in conjunction with decreases in snow cover fraction, and a statistically significant jump is identified in 1987 (p–value < 0.01) (Fig. 10d). However, although snow cover fraction is significantly lower throughout the entire melting season, only June and especially July show statistically significant increases in ER after 1987 (Fig. 11d). Therefore, the increase in potentially erosive rainfall, which is partially confirmed by recent observations (e.g. Meusburger et al., 2012), may only partially explain the rise in suspended sediment

concentration observed in July.

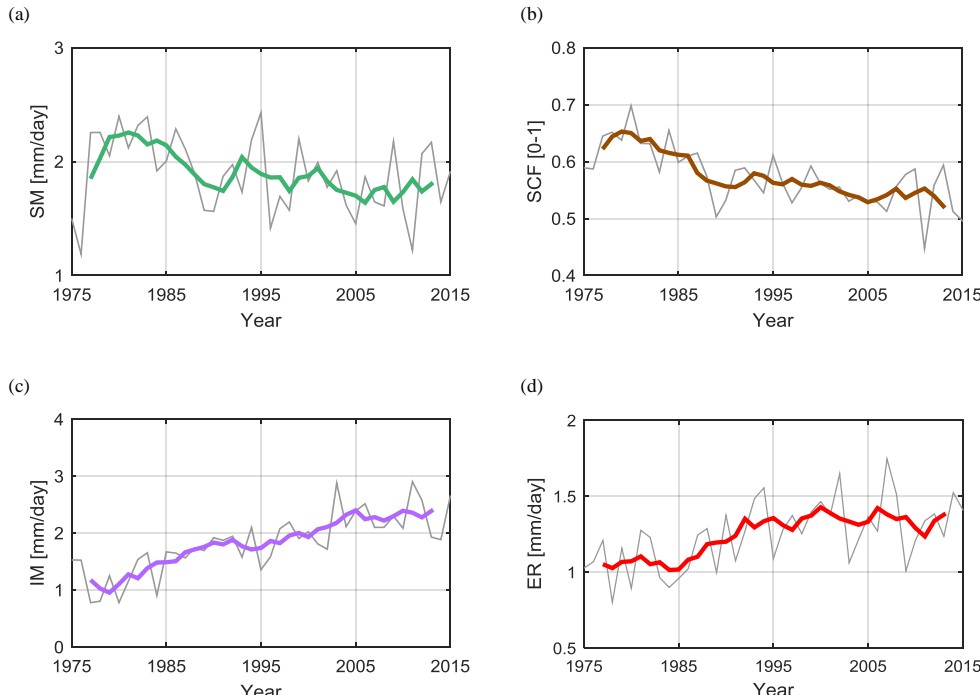

**Figure 10: Simulations for the period 1975–2015: (a) mean annual snowmelt SM; (b) mean annual snow cover fraction SCF; (c) mean annual ice–melt IM; (d) mean annual effective rainfall ER computed as rain falling over snow–free pixels. Mean annual values are shown in grey and a 5–year moving average is shown with a thick line.**

Third, our results show that temporal evolution of ice–melt is consistent with suspended sediment concentration rise. Although the change is rather gradual at the annual scale (Fig. 10c), the step–like increase in ice–melt is evident in the ice–melting season (May–September) and reaches highest magnitudes in July and August (Fig. 11c) in conjunction with rises in suspended sediment concentration in those months. The simultaneous increase in ice–melt and decrease in snowmelt

suggests that the abrupt warming has led to important alterations of the hydrological regime. To evaluate this alteration, we compute the relative contribution of rainfall, snow and ice–melt on the total annual sum of these three components. These contributions show a shift of snowmelt to ice-melt, which has accelerated in the mid–1980s (Fig. 12). The average relative contribution of ice–melt after 1987 increased by 10%, from 25.5% (1975–1986) to 35.5% in the following years (1987–2015). Particularly, from mid or late–1980s, the relative contribution of ice–melt has become greater than snowmelt,




indicating the substantial effect of the sharp increase in temperature on the basin hydrology. Enhanced ice–melt is coherent with the observed acceleration of Alpine glacier retreat after mid–1980s (see discussion).

Fluxes coming from glaciers are notoriously rich in sediments. Very fine silt–sized sediment resulting from glacier erosion is transported in suspension most often as wash load (Aas and Bogen, 1988). Proglacial areas generally represent rich sources
5  of sediment due to very active glacier erosive processes of abrasion, bed–rock fracturing and plucking (Boulton, 1974). Glacier retreat discloses large amount of sediments available to be transported by proglacial streams. Moreover, change in climatic conditions and specifically temperature–driven glacier recession and permafrost wasting may initiate specific erosional processes that consequently enhance sediment supply in proglacial environments (Micheletti et al., 2015; Micheletti and Lane, 2016; Lane et al. 2016). We conclude that the significant increase in ice–melt detected in the mid–
10  1980s (Fig. 10c, 11c, 12) is likely to be the main cause of the sharp rise in suspended sediment concentration entering Lake Geneva, through a combination of: (1) increased discharge originated in proglacial environments, which implies higher suspended sediment concentration; (2) larger relative contribution of sediment–rich ice–melt compared to snowmelt and precipitation fluxes; and (3) intensified sediment production and augmented sediment supply in proglacial areas due to rapid ice recession.

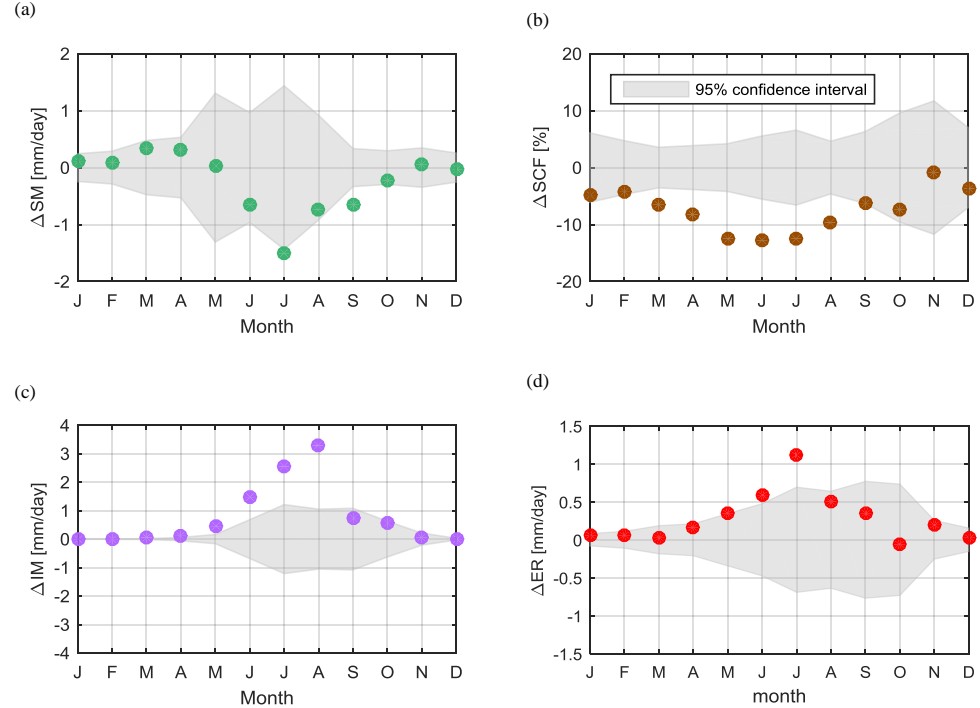

**Figure 11: Monthly differences between the period before and after the year–of–change (1975–1986 and 1987–2015): (a) mean snowmelt SM; (b) mean snow cover fraction SCF; (c) mean ice–melt IM; (d) mean effective rainfall ER computed as rain falling over snow–free pixels. Points outside the confidence interval (grey shaded area) represent statistically significant (5% significance level) changes in the monthly mean.**





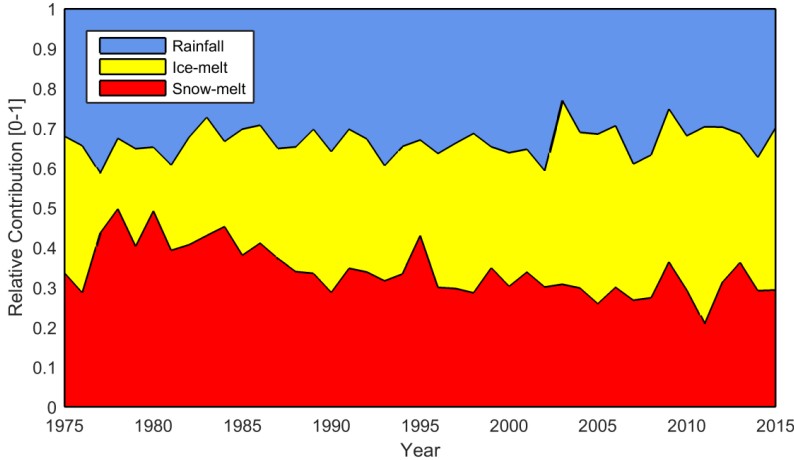

**Figure 12: Annual relative contribution of rainfall (R), ice–melt (IM) and snowmelt (SM) computed as the ratio between each component and their sum. Rainfall is extracted from observed precipitation by using a rain–snow temperature threshold, snow and ice–melt are simulated with spatially distributed temperature–index models.**

## 6. Discussion

### 6.1 Snow and Ice–melt Modelling

It is known that snowmelt factors in degree–day models vary with location, season, and time interval of temperature integration (Hock, 2003). Indeed, a wide range of values is reported in the literature (Hock, 2003). In the Swiss Alps, Schaefli et al. (2005) find an interval from 1.3 to 11.6 mm day$^{-1}$ °C$^{-1}$ for the snowmelt factor. Due to the simplicity of the degree–day approach, and considering all hydrological processes and the complex topography involved, a large variability in space should be expected. In the upper Rhône Basin, Boscarello et al. (2014) found a snowmelt factor equal to 4.3 mm day$^{-1}$ °C$^{-1}$ based on previous studies on the Toce Basin in Italy (Corbari et al., 2009). Calibration of a semi–lumped conceptual model for the three tributary catchments of the upper Rhône Basin – Lonza, Drance and Rhône at Gletsch – led to melt factors equal to 6.1, 4.5 and 6.6 mm day$^{-1}$ °C$^{-1}$, respectively (Schaefli et al. 2005). Thus, our calibrated value of $k_{\text{snow}} = 3.6$ mm day$^{-1}$ °C$^{-1}$, lies within reasonable ranges for this region. Differences in $k_{\text{snow}}$ between this and previous studies are attributable to different temporal resolution of models, lengths of calibration datasets, type and thresholds of precipitation partitioning, climatic inputs, threshold temperature for melt, and others.

Ice–melt factors also show large regional and temporal variability. A range from 5 to 20 mm day$^{-1}$ °C$^{-1}$ has been reported in the literature (e.g. Hock, 2003; Schaefli, 2005). As already discussed, this is partially due to the complexity of the snow and ice–melt processes in Alpine basins. Our calibrated value of $k_{\text{ice}} = 6.1$ mm day$^{-1}$ °C$^{-1}$ is in general agreement with previous studies carried out in this region (e.g. Schaefli et al., 2005; Boscarello et al., 2014). It should be noted that the simulated fluxes shown in Fig. 6 are upper estimates of runoff, because we are neglecting evaporation (evapotranspiration) in these high Alpine catchments. Evaporation plays indeed a secondary role in the long–term water balance in Alpine environments compared to precipitation and snowmelt (Braun et al., 1994; Huss et al., 2008). Considering that the aim of this study is to evaluate long–term changes in hydroclimatology and sediment dynamics of the upper Rhône Basin and not the short–term variability of ice–melt at the daily scale, we consider the snow and ice-melt model performance satisfactory. However, in





applications which require spatial distributions of snow and ice–melt we recommend to use approaches based on energy balance.

**6.2 Climate Impacts on Fine Sediment Dynamics**

In this paper we explored the link between climate and SSC measured at the outlet of the Rhône Basin, which represents an integrated measure of basin–wide fine sediment supply and transport. We demonstrated that warmer temperature could have resulted in higher SSC mainly through enhanced ice–melt and glacier retreat. It is notable that the sharp SSC rise can be detected at the basin outlet despite significant human impacts such as impoundments and flow abstraction. This demonstrates how even in highly human impacted and regulated catchments, a strong climatic signal in hydrological and sediment

dynamics can persist. This also suggests that the decrease in fine sediment load at the outlet of the upper Rhône Basin observed in the 1960s (Loizeau et al., 1997) could be the result of a combined effect of hydropower system development, as it has been hypothesized (Loizeau et al., 1997; Loizeau and Dominik, 2000), but also reduced ice–melt loads due to colder temperatures at the time. The cooling period, which occurred between 1950s and late 1970s (e.g. Beniston et al., 1994) was characterized by colder and snowy winters (e.g. Laternser and Schneebeli, 2003) and has been accompanied by reduced ice–

melt rates, glacier advance and positive glacier mass balances (Zemp et al., 2008; VAW–ETH, 2015).

The climate signal in sediment dynamics takes on particular importance in the context of climate change. Despite the large uncertainty, future projections under different climate change scenarios show a common tendency for Switzerland. A shift from a snow–dominated to a rain–dominated hydrological regime is projected by the end of the 21[st] century. Increase in winter precipitation, reduction of snowfall days, decrease in summer precipitation projected for the next century, result in a

reduced summer discharge and increased winter discharge. Snow cover is expected to disappear earlier at low elevations, while its duration and depth will be reduced at high elevations (Brönnimann et al., 2014; Bavay, 2009). Enhanced ice–melt in a warmer climate is expected to result in a massive reduction of ice volume of medium and large glaciers, while the majority of small glaciers will disappear in the next fifty years (Jouvet et al., 2011; Huss, 2016). Climate–induced alterations of the Alpine hydrological regime are expected to impact sediment dynamics considerably. The initial increase of washload

derived from enlarging proglacial areas may be followed by lower sediment export due to a reducing connectivity (Lane et al., 2016) and lower glacier mass. Changes in discharge distribution will alter transport capacity, enhancing or reducing the mobilization of coarser material. Furthermore, the changes in sediment yield will be spatially variable, because sediment sources contribute to total sediment yield depending on their distance to the outlet and their degree of sediment connectivity in space (Cavalli et al., 2013; Heckmann and Schwanghart, 2013; Bracken et al., 2015). In summary, predictions of

sediment fluxes are highly uncertain due to the complexity and feedbacks of the processes involved, inherent stochasticity in sediment mobilization and transport, and large regional variability in connectivity across the Alpine landscape. Although this paper focuses on past 40 years and does not look at future scenarios, it clearly shows that a more process–based approach, including the activation of sediment sources provide us with a better tool for analysing and attributing changes.

**6.3 Hydroclimatic Activation of Sediment Sources**

Our simulations show gradually decreasing snowmelt during the last forty years and significant shifts toward lower snow cover fraction in spring and summer in the Rhône Basin. These results are consistent with observations. The decreasing tendency in snow cover after mid or late 1980s has been demonstrated for the Swiss Alps (Beniston, 1997; Laternser and Schneebeli, 2003; Scherrer et al., 2004; Marty, 2008; Scherrer et al., 2006). Snow depth, number of snowfall days, and snow cover show similar patterns during the last century: a gradual increase until the early 1980s, interrupted in late 1950s and

early 1970s, and a statistically significant decrease afterwards (Beniston, 1997; Laternser and Schneebeli, 2003). Previous



analyses also state that the reduction in snow cover after mid–1980s is characterized more by an abrupt shift than by a gradual decrease (Marty, 2008), in agreement with our simulations. The reduction in snow cover duration, which is observed to be stronger at lower and mid altitudes than at higher elevations, is mainly the result of earlier snow melting in spring due to warmer temperatures (Beniston, 1997; Laternser and Schneebeli, 2003; Marty, 2008). Moreover, by analysing 76

meteorological stations in Switzerland, Serquet et al. (2011) demonstrated a sharp decline in snowfall days relative to precipitation days, both for winter and early spring, showing the impact of higher temperature on reduced snowfall, independently of variability in precipitation frequency and intensity. Therefore, despite the high complexity that characterizes snow dynamics in the Alps (Scherrer et al., 2006; 2013), the dominant effect of temperature rise on snow cover decline after late 1980s has been clearly shown (Beniston, 1997; Marty, 2008; Serquet et al. 2011; Scherrer et al., 2004;

2006). Both through earlier and larger snow melt in spring and lower snowfall in winter and early spring, the step–like temperature rise in mid–1980s has resulted in a significant decrease of snow cover and, as a consequence, snowmelt.

Reduced snow cover fraction implies both higher efficiency of ice–melting and larger susceptibility of the landscape to rainfall erosion. Accordingly, we found that the amount of liquid precipitation over snow free surfaces has significantly increased, which can play a significant role in increasing erosion and soil loss. Rainfall erosivity, expressed by the R–factor

of the Revised Soil Loss Equation (Wischmeier and Smith, 1978; Brown and Foster, 1987), was recently analysed for Switzerland. Although, the upper Rhône Basin together with the Eastern part of Switzerland was found to have relatively low rainfall erosivity (low R–factor) compared to the rest of the country due to a lower frequency of thunderstorms and convective events (Schmidt et al., 2016), there is evidence of an increasing trend for the R–factor from May to October during the last 22 years (1989–2010) (Meusburger et al., 2012). This reinforces our argument that the increase in effective

rainfall on snow–free surfaces may have contributed to suspended sediment concentration rise, through a combination of reduced snow cover fraction, increased rainfall–snowfall ratio and possible increases in rainfall intensity. However, simulations show a statistically significant jump in effective rainfall in June and July, while SSC is significantly larger in July and August. Therefore, we argue that erosive rainfall alone is unlikely to explain the abrupt jump in suspended sediment concentration observed around mid–1980s.

Conversely, we argue that SSC rise is mostly due to enhanced ice–melt and glacier retreat. The significant shifts towards higher ice–melt, reproduced in our simulations for all spring and summer months, are confirmed by observed accelerations of ablation rates, and glaciers' retreats occurred in mid–1980s in the European Alps. Ground–based and satellite observations, combined with mass balance analysis, reveal that current rates of glacier retreat are consistently greater than long–term averages (Paul, 2004; 2007; Haeberli, 2007). Estimations of glacier area reduction rates indicate a loss rate for the

period 1985–1999, which is seven times greater than the decadal loss rate for the period 1850–1973 (Paul, 2004). Investigations with satellite data and in–situ observations suggest that the volume loss of Alpine glaciers during the last thirty years is more attributable to a remarkable down–wasting rather than to a dynamic response to changed climatic conditions (Paul, 2004; 2007). Haeberli et al. (2007) estimated that glaciers in the European Alps lost about half their total volume (roughly 0.5% year$^{-1}$) between 1850–1975, another 25% (1% year$^{-1}$) between 1975–2000, and an additional 10–15%

(2–3% year$^{-1}$) in the period 2001–2005. The appearance of proglacial lakes and rock outcrops with lower albedo and high thermal inertia, separation of glaciers from the accumulation area, and general albedo lowering in European Alps (Paul, 2005), are among the main positive feedbacks that accelerate glacier disintegration and make it unlikely to stop in the near future (Paul, 2007). Although glacier dynamics are quite complex and involve many variables and feedbacks, the predominant role played by temperature rise in glacier wasting has been clearly demonstrated (e.g. Oerlemans, 2000). The

major volume loss in the recent past in Swiss Alpine glaciers is attributable to negative mass balances during the ablation season rather than to a lower accumulation by precipitation (Huss, 2008). For small high altitude Alpine glaciers, Micheletti and Lane (2016) showed negligible ice melt contributions to runoff between the mid–1960s and mid–1980s, after which contributions increased markedly.





Higher meltwater runoff in proglacial areas may evacuate larger amount of accumulated fine sediment. A larger relative contribution of sediment–rich meltwater results in larger SSC, and rapid ice recession enhances erosion and fine sediment supply in proglacial environments. While the rise in temperature can be detected for all spring and summer months, SSC has increased significantly only in July and August, which coincides with the greatest increases in simulated ice-melt. This

reinforces the argument for enhanced ice–melting being the main reason for higher fine sediment concentrations at the outlet of the catchment.

## 7. Conclusions

The aim of this research was to analyze changes in the hydroclimatic and suspended sediment regime of the upper Rhône Basin during the period 1975-2015. We show an abrupt increase in basin-wide mean air temperature that occurred in the

mid–1980s. The simultaneous step–like increase in suspended sediment concentration SSC at the outlet of the catchment, detected in July and August, suggests a causal link between fine sediment dynamics and climatic conditions. Two main factors link warmer climate and enhanced SSC: increased transport capacity and increased sediment supply resulting from spatial and/or temporal activation-deactivation of sediment sources. Our results show that transport capacity through discharge is not likely to explain the increases in SSC, because no statistically significant changes in the mid–1980s are

present in Rhône Basin discharge, neither at the annual nor monthly timescales. The suggestion is that the impact of warmer climatic conditions acts on fine sediment dynamics through the activation and deactivation of different sediment sources and processes.

To understand sediment supply conditions we analyzed the temporal evolution of three main sediment sources: (1) sediments sourced and transported by snowmelt along hillslopes and channels; (2) mass wasting events, hillslope and channel bank

erosion driven by erosive rainfall events over snow–free surfaces; (3) fine sediment flux generated by glacier–related ice–melt. The fluxes of snow and ice–melt together with snow cover fraction and rainfall were analyzed to detect changes in time and their coherence with changes in SSC. The simulations of daily snowmelt, ice–melt and snow cover fraction for the last forty years were generated by a snow and ice–melt model, and potentially erosive rainfall was computed as the liquid precipitation falling on snow–free cells.

Our results show that while mean annual precipitation does not show any evident change between the periods before and after the SSC jump in mid–1980s, potentially erosive rainfall clearly increases over time especially in June and July, but not August. On the other hand, ice–melt has significantly increased due to temperature–driven enhanced ablation. Statistically significant shifts in ice–melt were identified for summer, with highest increases in July and August, in accordance with the rise in SSC. Concurrently to the temperature and SSC change, the relative contribution of ice–melt to total annual runoff

(sum of rainfall, snow and ice–melt) presents a significant increase in mid–1980s, shifting the hydrological regime of the Rhône Basin from snowmelt dominated to ice–melt dominated. Based on these results we propose that climate has an effect on fine sediment dynamics by the activation and deactivation of the three main sediment sources in the Rhône Basin, and that ice–melt plays a dominant role in the suspended sediment concentration rise in the mid–1980s through: (1) increased flow derived from sediment–rich proglacial areas; (2) larger relative contribution of sediment–rich ice–melt compared to

snowmelt and precipitation; and (3) increased sediment supply in hydrologically connected proglacial areas due to glacier recession. While snowmelt has decreased, the reduced extent and duration of snow cover may also have contributed to the suspended sediment concentration rise through enhanced erosion by heavy rainfall events over snow free surfaces.

Because changes in SSC are not consistent with changes in discharge and transport capacity, our work emphasizes how the inclusion of sediment sources and their activation in catchments is necessary for attributing change. This is particularly

important when climate change impact assessments and projections are made, and when hydropower operation impacts are





being quantified. The latter is a crucial point in the Rhône Basin where sediment fluxes are affected by flow regulation due to hydropower production and by grain–size dependent trapping in reservoirs.

**Author contribution**

A. Costa and P. Molnar designed the methodology. A. Costa developed the code and carried out simulations and computations. A. Costa prepared the manuscript with contributions from all co-authors.

The authors declare that they have no conflict of interest.

**Acknowledgements**

We thank Christoph Frei (Federal Office of Meteorology and Climatology MeteoSwiss) for providing us with experimental temperature and precipitation datasets and for suggestions on the right use of Meteoswiss gridded data and the application of statistical tests. The Federal Office of the Environment (FOEN) provided discharge and suspended sediment concentration data. We thank Alessandro Grasso (FOEN) for the explanation of the SSC data collection procedures. This research was supported by the Swiss National Science Foundation Sinergia grant 147689 (SEDFATE).





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
