# Peer review of "Temperature signal in suspended sediment export from an Alpine catchment"

_Hydrology and Earth System Sciences, 2017_

## Referee Comment (RC1) · Anonymous Referee #1 · 7 Feb 2017

General comments

This manuscript focuses on the role of climatic forcing in sediment production and transport in a large Alpine catchment. It applies a degree-day model to explain changes in suspended sediment concentrations resulting from hydro-climatic activation of sediment sources within the study catchment. This represents an interesting approach that has been implemented with consideration of processes influencing rates of sediment supply from the range of catchment sediment sources. The manuscript is well written and describes clearly the modelling approach. There is scope for potential re-structuring of the manuscript, which contains some repetition and is over-long in places. Nonetheless, such changes amount to only a minor level. Overall, this is a technically sound and interesting study that deserves publication.

[Figure]

Specific comments

Lines 20-35, page 2: Description of catchment sediment sources. Given the number of figures in the manuscript, I suggest the authors remove the overly simplistic schematic in Fig. 1 as it adds little beyond that which is available from the summary in the text.

Lines 20-25, page 4: The discussion of anthropogenic impacts in the catchment mentions gravel mining operations along the main channel and tributaries. Such direct disturbance of the channel could increase suspended sediment supply. Can this effect be discounted completely as a factor contributing to the observed trend in suspended sediment concentration (Fig 7c)?

Lines 10-15, page 6: Table 1 summarises some of the information given in Section 4 Data Description. Moreover, Section 3.3 Calibration and Validation also contains some description of the datasets used. To reduce repetition, can section 4 be shortened or consolidated? Perhaps a shortened descriptive summary of the datasets could be provided alongside Table 1 before introducing the models.

Lines 30-35, page 10: Could the use of fixed interval sampling (twice per week) for suspended sediment concentrations (SSC) influence the interpretation of trends during the observation period? The absence of continuous measurements (i.e. turbidity) or stage-triggered sampling may result in under-estimation of mean annual SSC because elevated but short-duration peaks in flow and SSC are less likely to be captured by fixed interval sampling. For this reason, the potential effect of the reported increase in direct rainfall on snow-free surfaces ('effective rainfall') on SSC could be underestimated because such events are short duration compared to the longer duration ice and snowmelt effect on SSC. This deserves consideration when evaluating the relative contributions of rainfall, snow and ice-melt (page 18) to observed trends in SSC.

Sections 5 & 6: I recommend merging the Results and Discussion. The Results section contains some elements of discussion (e.g. lines 20-25, page 15, on climate patterns), while in multiple locations within the Results section the authors write 'see discussion'.

[Figure]

The integration of Results and Discussion could produce a more coherent paper that presents findings and their interpretation in relevant sub-sections. For example, the discussion of snow and ice-melt modelling in terms of previously reported melt factors (lines 10-25, page 20) would fit logically with the presentation of the calibration results (section 5.1).

Section 6.2 (lines 15-35, page 21): The discussion of future climate change effects on the sediment regime should be shortened and focus mainly on the point about the value of a more process-based approach. The paper offers no evidence based on future change simulations, so should limit speculative discussion in this area.

---

## Author Comment (AC1) · 15 Feb 2017

We thank Referee #1 for his helpful review. We have analysed her/his suggestions and we report in the following our response to each specific comment.

1. Lines 20-35, page 2: Description of catchment sediment sources. Given the number of figures in the manuscript, I suggest the authors remove the overly simplistic schematic in Fig. 1 as it adds little beyond that which is available from the summary in the text.

We agree with Referee #1 and we will remove Fig. 1 from the revised manuscript.

2. Lines 20-25, page 4: The discussion of anthropogenic impacts in the catchment mentions gravel mining operations along the main channel and tributaries. Such direct

disturbance of the channel could increase suspended sediment supply. Can this effect be discounted completely as a factor contributing to the observed trend in suspended sediment concentration (Fig 7c)?

We are aware that, for short periods after river bed disturbance, gravel extraction may cause local releases of fine sediments from the river bed. However, this process is unlikely to affect the suspended sediment load and balance at the outlet of the basin over seasonal and annual timescales. This is confirmed by the volume of gravel extracted along the main Rhône River and along tributaries, available from 1989. Annual volumes of gravel extracted, expressed as difference from the average over the period 1989-2014, do not show any clear correlation with mean annual suspended sediment concentration (Fig. 1). In agreement with Fig. 1, the coefficient of determination between the two variables is very low (R2 = 0.08). Therefore, we conclude that, although gravel mining data for comparing the periods before and after mid-1980s are not available, gravel extraction is unlikely to play a significant role in the suspended sediment concentration rise observed in mid-1980s.

3. Lines 10-15, page 6: Table 1 summarises some of the information given in Section 4 Data Description. Moreover, Section 3.3 Calibration and Validation also contains some description of the datasets used. To reduce repetition, can section 4 be shortened or consolidated? Perhaps a shortened descriptive summary of the datasets could be provided alongside Table 1 before introducing the models.

We agree with Referee #1. We will maintain Section 4 but we will reduce it. Particularly, we will shorten data description in Section 4.1 (Precipitation and Air Temperature) and Section 4.2 (Discharge and Suspended Sediment Concentration).

4. Lines 30-35, page 10: Could the use of fixed interval sampling (twice per week) for suspended sediment concentrations (SSC) influence the interpretation of trends during the observation period? The absence of continuous measurements (i.e. turbidity) or stage-triggered sampling may result in under-estimation of mean annual SSC because

elevated but short-duration peaks in flow and SSC are less likely to be captured by fixed interval sampling. For this reason, the potential effect of the reported increase in direct rainfall on snow-free surfaces ('effective rainfall') on SSC could be underestimated because such events are short duration compared to the longer duration ice and snowmelt effect on SSC. This deserves consideration when evaluating the relative contributions of rainfall, snow and ice-melt (page 18) to observed trends in SSC.

We thank Referee #1 for this constructive comment. We agree that discontinuity in the sampling procedure may indeed influence the interpretation of trends in suspended sediment concentration, and we will discuss this in the revised manuscript. However, although it is true that the manual suspended sediment sampling is not conducted on extreme flood days, the sampling is representative of the probability distribution of daily streamflow – we will show this in the final response to the reviewers and the revised manuscript. We have also redone all our analysis of changes in annual ER, SM, IM based on rates computed only on days when suspended sediment was sampled and we find that the results are unchanged. The figures supporting this will also be added in the final response to the reviewers and the revised manuscript.

5. Sections 5 & 6: I recommend merging the Results and Discussion. The Results section contains some elements of discussion (e.g. lines 20-25, page 15, on climate patterns), while in multiple locations within the Results section the authors write 'see discussion'. The integration of Results and Discussion could produce a more coherent paper that presents findings and their interpretation in relevant sub-sections. For example, the discussion of snow and ice-melt modelling in terms of previously reported melt factors (lines 10-25, page 20) would fit logically with the presentation of the calibration results (section 5.1).

We agree that sections of results and discussion can be organized in a more coherent structure. In the revised manuscript, we will move the discussion on snow and ice-melt modelling (Section 6.1) to the section on models calibration (Section 5.1). In addition, Section 6.2 and 6.3 will be merged and reduced, maintaining only the most significant

discussion points.

6. Section 6.2 (lines 15-35, page 21): The discussion of future climate change effects on the sediment regime should be shortened and focus mainly on the point about the value of a more process-based approach. The paper offers no evidence based on future change simulations, so should limit speculative discussion in this area.

We agree that the discussion on climate change projections in Switzerland of Section 6.2 is too extended. However, as suggested by Referee #1, we would like to focus on the importance of adopting a more process-based approach when analysing the effects of climate change on suspended sediments. Therefore, we will reduce the discussion and we will merge Section 6.2 with Section 6.3. By linking these two sections, we aim at emphasizing that looking at erosional and transport processes driven by different hydroclimatic variables may allow the quantification of the effects of changes in climatic conditions on the sediment regime.

———————————

[Figure]

**Fig. 1.** Fig. 1: Mean annual suspended sediment concentration (SSC) and total annual volume of gravel extracted, expressed as difference from the average over the period 1989-2014

---

## Referee Comment (RC2) · Anonymous Referee #2 · 20 Feb 2017

In this manuscript the increase in suspended sediments observed in the Rhone river after 1987 (compared to the period 1960-198) is statistically compared to hydro-climatic factors as temperature, rainfall, discharge, snow cover, snow melt and ice melt. Interpolated meteorological products and satellite observation of snow cover were used to model snow cover, snow and ice melt for the Rhone basin with degree-day melt models. The meteorological and the modelled data were then statistically compared to measured discharge and measured sediment supply. From this comparison, the authors suggest that the observed changes in the suspended sediment concentration is mainly due on the one hand to a reduced extent and duration of snow cover leading to larger areas exposed to erosive rainfall and on the other hand to an increase in ice melt in the summer months.

This study represents a valuable contribution for understanding the impact of climatic

data on sediment transport. The presented approach is interesting and innovative and the results and analysis are meaningful. The study is also well-documented. Overall the manuscript is well written but substantial improvement can be made on the structure of the paper. This would avoid many repetitions throughout the text. Especially the discussion part needs to be built more on the results of the studies itself to be less speculative (see main comments below). Furthermore, I have some concern on the assumption to not model glacier retreat in the ice model, as this can lead to over-estimation of the ice melt. There is therefore a need to further discuss the impact of the simplicity of the ice model on the results (see main comments below). The issues presented in this manuscript are in the scope of HESS and as my comments concern substantial restructuration of the manuscript and further development in the analysis of the ice model results, I would support its publication after major revisions.

Main comments :

Goal formulation (p. 3, l. 22-25): One of the goals of the study (aim c) is to assess the future climate changes and hydropower operation impacts. This is only discussed later in the discussion, but only in a speculative way as the analysis itself does not include any hydropower data or future climate scenario. As it is formulated in the aims of the study, I would expect future climate scenario being taken into account and hydropower operation being explicitly analyzed. Instead the observations on this impact are only speculative (see for example discussion p. 21, l. 16-33). I would therefore suggest the authors to reformulate their study aims and also to minimize the weight they give to this topic in the discussion as it is only speculative.

Overall structure:

The overall structure of the paper leads to many repetitions throughout the text.

1. The main problem is that parts of the discussion can be found in the methods, results, and discussion sections. This makes the text repetitive and the information hard to find. The authors should define a clear structure for their manuscript. They

should decide if they want the results and discussion together in one section, meaning a discussion following each results description, or if they want 2 distinct sections (which I personally would recommend), meaning that the result section only describe the results and all discussion and interpretation of the results is moved to discussion section.

2. The data are described first in the method section and then in the data description section. This is redundant. Best would be to shortly describe the data before the method section.

Section 3.1, snow model:

Several parameters that play an important role in the degree day approach (as SD, Trs, Tsm) are set before calibration. A sensitivity analysis of the impacts of these parameters would be interesting to assess their impact on the results and give them more credit.

Ice model:

In my opinion the most critical point of the method is the simplicity of the ice melt simulation. The ice model uses a simple degree day routine and do not considers changes in glacier areas, which might be consequent over a time period of ca 50 years. The authors claim "Temporal dynamics of glacier coverage are not accounted for —- ice accumulation, glacier retreat and ice movement are disregarded. The reduction of Alpine glaciers for the period 1950–2000 was estimated to be within the 10 range 500–1000 m (Hoelzle, 2003; Oerlemans, 2005), while our effective climate grid resolution is 2×2 km, i.e. the retreat is considerably lower than the grid resolution of climatic inputs. The consideration of ice dynamics would therefore add a degree of complexity that our spatial resolution cannot take advantage of" (p. 11, l. 8-12). I disagree with this argumentation. The grid size of the meteorological product is surely important for the modelling of glacier retreat, but many studies used these meteorological dataset for modelling changes in glacier mass balance in the Alps and proved that the results

were accurate enough. The model resolution for the ice melt calculation is 250 m and is therefore "small" enough to be influenced by a glacier retreat in range of 500-1000 m. My concern is that if the glacier retreat is not taken into account in the modelling of such a long period (50 years, with very high rates of glacier retreat in the last decade), there is a real risk of overestimating the glacier melt by the model. The results of the analysis suggest that there is a shift in the discharge from a snow-melt dominated regime to an ice-melt dominated regime (Fig. 10 and 12). I wonder if part of this increase in ice-melt can be induced by the fact that glacier retreat is not taken into account. Therefore I would suggest the authors to discuss in more details the impact of this model assumption on the results. Many glaciers in the Rhone basin are well-documented and have yearly mass balance data. It would be worth to compare the modeled ice melt values with measured time series of glacier mass balance to exclude that the increase of ice melt is due to model assumptions.

Detailed comments:

p.2, l. 13-18: " In such…": this sentence is too long, make 2 sentences out of it. Isn't rainfall and liquid precipitation the same?

P.3, l. 25-27: "Although our results…": move to the discussion.

p.4, Fig. 1: In my opinion Fig. 1 does not give any additional information. The aim of this figure is not very clear. I would suggest to suppress it or to develop it in a way where it is clear how the 4 sediment sources are build and how they can play a role. Maybe the authors could also link the different sources to the hydro-climatic factors (which factor influences which source).

p.5, l. 1-2: " However … of our investigation": This does not belong to the description of the study site, it belongs rather to the introduction or to the discussion.

p.5, l. 9-11: which are the observed and the modeled data?

p.6, l. 1-3: how do the authors calculate the 250-250m daily temperature and precipitation from the Meteoswiss gridded dataset (ca. 2x2km)? It is not clear to me if or when you use monthly average over the basin or daily gridded data. Please clarify.

p.6, Table 1: the gridded dataset from Meteoswiss is originally in degree; therefore the grid size is ca 2x2km.

p.8, l. 4-10: this belongs to the discussion (see also comments above on the ice model). This is also repeated in p.11, l. 8-12.

p.9, l. 6-18: Most of this paragraph belongs to the description of the study area, not to the methods section. p.10, l. 18-21: " we also verified..." And? Where are the results of this analysis? What are the conclusions? I don't understand the sense of these sentences.

p. 10, l. 23-26: this is a repetition.

p.11, l.20: " This value...", it is enough to discuss it only in the discussion, delete this sentence.

p.11, l.31-32: belongs to the discussion.

p.12, l.1: " we are confident" avoid this kind of expression (all over the text) as it gives the impression that the observation are not based on results but on feelings.

p.12, l.3-5: "Despite...", it is enough to discuss it only in the discussion, delete this sentence.

Fig. 3a-b: I am not convinced that these 2 figures bring additional information. I would suggest deleting them.

p.15, l. 5-8; l.9-11: Move to the discussion.

p.15, l.16-24: In this paragraph it is confusing when annual and monthly discharge is meant.

p.15, l.22: "this variability is reflected in streamflow", where?

p.16, l.1-5: move to the discussion.

p.16, l.6-10: this part belongs to the discussion and is a repetition.

p.16, Fig.8: ΔQ and ΔP seems to react inversely, is there an explanation? Is it due to hydropower?

p.17, l.2: "The fact that sediment supply impacts suspended sediment concentration is evident..." really? Where do I see that?

p.17, Fig. 9: would it be possible to highlight (different colors) < 1987 and >1987?

p.17, l.27: " Our simulation...", it is enough to discuss it only in the discussion, delete this sentence.

p.17, l.28-30: How did you come to this conclusion? Did you compare it with literature? These sentences belong to the discussion. p.18, l.2-6; 8-10: move to the discussion.

p.18, l. 16: where is this shown? Cite the figure.

p.18, l.22: " a shift of snow-melt to ice-melt": I would not call it a shift as it is only slightly visible from fig. 12. Is it statistically significant? Cite the figure where the shift can be seen (fig. 12?).

p.19, l.1-15: Most of this belongs to the discussion.

p.20, l. 20-23: this has already been said and is only a repetition.

p.21, l.10: "...decrease in fine sediment load..." where is this decrease in the 60s to see? The analysis start around 1965, where the sediment load is low, but there is no decrease...

p.21, l. 10-12: how do you come to the conclusion that it can be an effect of hydropower?

p.21, l.16-34: This is only speculative and take too much importance in the discussion.

p.22, l.8-10: repetition

p.22, l.14-19: sentence too long, shorten it or separate it.

p.22, l.21: there is no increase in rainfall intensity in the analysis (fig. 7)

p.23, l.18-21: There were 4 sources of sediments in fig. 1, please clarify. To me it is not clear how these sediment sources were taken into account in the analysis. You should maybe emphasis in the discussion how each factor influences each sediment source.

―――――――――――――――――――

---

## Author Comment (AC2) · 1 Mar 2017

We thank Referee #2 for his/her review. We consider that the comments on snow (comment "Section 3.1, snow model") and ice model (comment "Ice model") are the most important. Therefore, we discuss here these two points, while other comments on goals, structure, and more detailed suggestions will be implemented in the revised manuscript. To address the concerns of the Referee about the ice-melt component of our work, we performed additional analyses and outline the results here. This discussion and results will also be reproduced in the revised manuscript.

1- Referee's Comment on Section 3.1, snow model: Several parameters that play an important role in the degree day approach (as SD, Trs, Tsm) are set before calibration. A sensitivity analysis of the impacts of these parameters would be interesting to assess

their impact on the results and give them more credit.

1- Authors Reply on Section 3.1, snow model: We agree with Referee #2 that it would be interesting to estimate the impact of parameters on the snow model results. Therefore, we will perform a sensitivity analysis on TRS, TSM and Ksnow as the key parameters of the snow model. We will also assume different snow depth thresholds for considering what is a snow covered surface. Results will be included in the revised manuscript.

2- Referee's Comment on Ice model: In my opinion the most critical point of the method is the simplicity of the ice melt simulation. The ice model uses a simple degree day routine and do not considers changes in glacier areas, which might be consequent over a time period of ca 50 years. The authors claim "Temporal dynamics of glacier coverage are not accounted for — ice accumulation, glacier retreat and ice movement are disregarded. The reduction of Alpine glaciers for the period 1950–2000 was estimated to be within the 10 range 500– 1000 m (Hoelzle, 2003; Oerlemans, 2005), while our effective climate grid resolution is 2_2 km, i.e. the retreat is considerably lower than the grid resolution of climatic inputs. The consideration of ice dynamics would therefore add a degree of complexity that our spatial resolution cannot take advantage of" (p. 11, l. 8-12). I disagree with this argumentation. The grid size of the meteorological product is surely important for the modelling of glacier retreat, but many studies used these meteorological dataset for modelling changes in glacier mass balance in the Alps and proved that the results were accurate enough. The model resolution for the ice melt calculation is 250 m and is therefore "small" enough to be influenced by a glacier retreat in range of 500-1000 m. My concern is that if the glacier retreat is not taken into account in the modelling of such a long period (50 years, with very high rates of glacier retreat in the last decade), there is a real risk of overestimating the glacier melt by the model. The results of the analysis suggest that there is a shift in the discharge from a snow-melt dominated regime to an ice-melt dominated regime (Fig. 10 and 12). I wonder if part of this increase in ice-melt can be induced by the fact that glacier retreat is not taken into

account. Therefore I would suggest the authors to discuss in more details the impact of this model assumption on the results. Many glaciers in the Rhone basin are well documented and have yearly mass balance data. It would be worth to compare the modeled ice melt values with measured time series of glacier mass balance to exclude that the increase of ice melt is due to model assumptions.

2- Authors Reply on Ice model: This is indeed a very important point, and we agree with Referee #2 that neglecting glacier dynamics may influence the estimation of ice-melt rates. The Referee raises the possibility that by neglecting glacier volume loss (retreat) we are possibly overestimating the ice-melt contribution over our study period. To provide evidence that this is not the case, we compared our simulations with time series produced with the Global Glacier Evolution Model (GloGEM), a model accounting both for the main mass balance components and glacier dynamics. For comparison, we used total monthly runoff (snowmelt + ice-melt + rainfall) generated at the glaciated surfaces of the upper Rhone basin, simulated with GloGEM (Huss and Hock, 2015) for the period 1980-2010.

GloGEM computes the mass balance for every 10-m elevation band of each glacier, by estimating snow accumulation, snow and ice melt, and refreezing of rain and melt water. The response of glaciers to changes in mass balance is modelled on the basis of an empirical equation between ice thickness changes and normalized elevation range parametrized as proposed by Huss (Huss et al., 2010). Normalized surface elevation changes $\Delta hr$ are derived for each elevation band from mass balance changes (mass conservation). Starting from initial values derived by the method of Huss and Farinotti (2012), ice thickness is updated at the end of each hydrological year by applying the relation between normalized elevation range $hr$ and normalized surface elevation change $\Delta hr$. The area of each glacier is finally adjusted by a parabolic cross-sectional shape of the glacier bed (Huss and Hock, 2015). GloGEM is calibrated and validated over the period 1980-2010 with estimates of glacier mass changes by Gardner et al. (2013) and in situ measurements provided by the World Glacier Monitoring Service.

Although in our hydrological model, which considers precipitation, snowmelt, ice-melt at pixel scale, but integrates them to basin-average values, we do not include glacier dynamics, the total annual volumes of runoff (snowmelt + ice-melt + rainfall) from glaciated areas, correlate very well with results of GloGEM (Fig. 1a). Measures of performance confirm the agreement between the two models: the correlation coefficient is equal to 0.86 and the Nash-Sutcliffe efficiency is equal to 0.67. We are also capable to capture quite well the seasonal pattern of runoff generated from glaciated areas (Fig. 1b).

Perhaps most importantly, GloGEM simulations show that total annual runoff is increasing throughout the period and there is no evidence for a drop in ice-melt rates. This confirms that, although glaciers of the upper Rhone basin are retreating, sediment-rich fluxes originated at glacial and proglacial areas are increasing during the 1980-2010 period. As expected, total runoff from glaciated surfaces and ice-melt are highly correlated (Fig. 1a). In our model, the correlation coefficient between the two variables is equal to 0.95. Therefore, the increasing tendency of total runoff simulated with GloGEM indicates that ice-melt component is most likely also rising. Non-parametric Mann-Kendall tests indicate an increasing trend with 5% significant level for total runoff and ice-melt simulated with our model and for total runoff simulated with GloGEM. Trend slopes, estimated with the Theil-Sen estimator, confirm that most likely we are not overestimating the rate of increase in ice-melt. Indeed, we find ~27.65 mio m3/year and ~21.71 mio m3/year, respectively for total runoff simulated with GloGEM and with our model, and only ~17.90 mio m3/year for ice-melt simulated with our model. We also computed the basin-averaged mass balance accounting for snow accumulation and snow and ice-melt for each hydrological year. The mean mass balance rate over the period1980-2010 is equal to -0.78 $\pm$ 0.22 m w.e./year (Fig 2). This value is slightly greater than that found by Fischer et al. (2015) for the upper Rhone basin (-0.59 m w.e./year ), but within the uncertainty of the estimate. In summary, we are confident that we can state that, although we do not account for glaciers retreat, our model results agree with a much more complex physical-based modelling approach including

glacier dynamics. Both comparisons with GloGEM and our basin-averaged mass balance indicate that we are not significantly overestimating ice-melt contribution during the period 1975-2015.

References

Fischer, M., Huss, M., Hoelzle, M.: Surface elevation and mass changes of all Swiss glaciers1980–2010, Cryosphere 9, 525–540, doi:10.5194/tc-9-525- 2015, 2015.

Gardner, A. S., Moholdt, G., Cogley, J. G., Wouters, B., Arendt, A. A.,Wahr, J.: A reconciled estimate of glacier contributions to sea level rise: 2003 to 2009, Science, 340,852–857, doi:10.1126/science.1234532, 2013.

Huss, M. and Farinotti, D.: Distributed ice thickness and volume of all glaciers around the globe, J. Geophys. Res., 117: F04010, doi: 10.1029/2012JF002523, 2012.

Huss, M. and Hock, R.: A new model for global glacier change and sea-level rise, Front. Earth Sci., 3:54, doi: 10.3389/feart.2015.00054, 2015.

Huss, M., Jouvet, G., Farinotti, D., Bauder, A.: Future high-mountain hydrology: a new parameterization of glacier retreat, Hydrol. Earth Syst. Sci., 14, 815–829, doi:10.5194/hess-14-815-2010, 2010.

[Figure]

**Fig. 1.** Runoff (snowmelt + ice-melt + rainfall) generated at glaciated areas within the upper Rhone basin, simulated with GloGEM and with our temperature index model (TI) for the period 1980-2010: (a)

**Fig. 2.** Mass balance rate for glaciated areas of the upper Rhone basin, simulated with our temperature index model for the period 1975-2015.

---

## Author Response (AR1)

**Reply to reviewers [paper hess−2017−2]**

**Temperature signal in suspended sediment export from an Alpine catchment**

Anna Costa, Peter Molnar, Laura Stutenbecker, Maarten Bakker, Tiago A. Silva, Fritz Schlunegger, Stuart N. Lane, Jean–Luc Loizeau, Stéphanie Girardclos

We thank the reviewers and the Editor for their helpful reviews. We have analysed their suggestions and we have revised the manuscript accordingly. We report in the following our response to each specific comment.

**Editor**

*Dear authors, following up on the reviews received for your manuscript it appears that both referees rate your contribution as highly interesting and relevant for the HESS reader community. The two referees (and I do agree) see much potential for increasing the readability and structure of your paper – please follow their quite detailed suggestions for improvement in this respect. One referee sees some need for clarification related to the conceptualization and results of the ice model (essentially due to the fact that no glacier retreat is modelled). You may also consider to develop on this more specific aspect in your revised version.*

We thank the Editor for the feedbacks. We have edited the manuscript to increase its readability. We have followed the suggestions of the reviewers and we have added to the revised manuscript results related to the following four main aspects:
(1) sensitivity analysis of our snowmelt model parameters;
(2) quantification of the possible overestimation of icemelt rates due to fact that glacier retreat is not modelled;
(3) analysis of the effects of discontinuous SSC sampling on trend and jumps detected during the observation period;
(4) estimation of potential impacts of gravel mining activities on SSC rise detected in mid−1980s.

**Anonymous Referee #1**

*General comments*
*This manuscript focuses on the role of climatic forcing in sediment production and transport in a large Alpine catchment. It applies a degree−day model to explain changes in suspended sediment concentrations resulting from hydro−climatic activation of sediment sources within the study catchment. This represents an interesting approach that has been implemented with consideration of processes influencing rates of sediment supply from the range of catchment sediment sources. The manuscript is well written and describes clearly the modelling approach. There is scope for potential re−structuring of the manuscript, which contains some repetition and is over−long in places. Nonetheless, such changes amount to only a minor level. Overall, this is a technically sound and interesting study that deserves publication.*

We thank Referee #1 for this helpful review. We have analysed her/his suggestions and we report in the following our response.

*Specific comments*
*1) Lines 20−35, page 2: Description of catchment sediment sources. Given the number of figures in the manuscript, I suggest the authors remove the overly simplistic schematic in Fig. 1 as it adds little beyond that which is available from the summary in the text.*

1) We agree with Referee #1 and we removed Fig. 1 from the revised manuscript.

*2) Lines 20−25, page 4: The discussion of anthropogenic impacts in the catchment mentions gravel mining operations along the main channel and tributaries. Such direct disturbance of the channel could increase suspended sediment supply. Can this effect be discounted completely as a factor contributing to the observed trend in suspended sediment concentration (Fig 7c)?*

2) We are aware that, for short periods after river bed disturbance, gravel extraction may cause local releases of fine sediment from the river bed. However, this process is unlikely to affect the suspended sediment load and balance at the outlet of the basin over seasonal and annual timescales. This is confirmed by the volume of gravel extracted along the main Rhone River and along tributaries, available from 1989. Annual volumes of gravel extracted, expressed as the difference from the average over the period 1989−2014, do not show any clear correlation with mean annual suspended sediment concentration

(Fig. 1). In agreement with Fig. 1, the coefficient of determination between the two variables is very low ($R^2 = 0.08$). Therefore, we conclude that, although gravel mining data for comparing the periods before and after mid−1980s are not available, gravel extraction is unlikely to play a significant role in the suspended sediment concentration rise observed in mid−1980s.

In the revised manuscript, we report these comments in Sect. 6.2 where we discuss other anthropogenic factors that could have potentially affected SSC dynamics but most likely did not play a relevant role in the SSC rise occurred in mid−1980s (i.e. river channelization, hydropower reservoirs).

[Figure]

**Fig. 1: Mean annual suspended sediment concentration (SSC) and total annual volume of gravel extracted, expressed as difference from the average over the period 1989−2014.**

*3) Lines 10−15, page 6: Table 1 summarises some of the information given in Section 4 Data Description. Moreover, Section 3.3 Calibration and Validation also contains some description of the datasets used. To reduce repetition, can section 4 be shortened or consolidated? Perhaps a shortened descriptive summary of the datasets could be provided alongside Table 1 before introducing the models.*

3) We agree with Referee #1. We reduced Sect. 4 and we changed Table 1, which now contains the list of the analysed variables, their source and the spatial and temporal resolution adopted in this analysis.

*4) Lines 30−35, page 10: Could the use of fixed interval sampling (twice per week) for suspended sediment concentrations (SSC) influence the interpretation of trends during the observation period? The absence of continuous measurements (i.e. turbidity) or stage−triggered sampling may result in under−estimation of mean annual SSC because elevated but short−duration peaks in flow and SSC are less likely to be captured by fixed interval sampling. For this reason, the potential effect of the reported increase in direct rainfall on snow−free surfaces ('effective rainfall') on SSC could be underestimated because such events are short duration compared to the longer duration ice and snowmelt effect on SSC. This deserves consideration when evaluating the relative contributions of rainfall, snow and icemelt (page 18) to observed trends in SSC.*

4) We thank Referee #1 for this constructive comment. We agree that discontinuity in the sampling procedure may indeed influence the interpretation of trends in suspended sediment concentration, and we agreed that this point deserved further investigation. Therefore, we estimate this potential effect by considering total daily, basin−averaged values of the hydroclimatic variables SM, IM, ER and Q only on days corresponding to SSC−measurement days. We compare these new time series ("SSC−measurement days") with the original ones ("all days") by computing the cumulative distribution functions and by testing the equality of mean monthly and annual values before and after mid−1980s. We consider only values greater than zero when computing the cumulative distribution function, to avoid the influence of many zeros. Although extremely high and low values are missed by the sampling method, cumulative distribution functions of SM, IM, ER and Q on SSC−measurement days and on all days result very similar (Fig. 2). This indicates that, although measurements of SSC are collected at a fixed interval, the sampling is representative of the process. In addition, results of the statistical tests on mean monthly and mean annual values of all analyzed hydroclimatic variables are unchanged. This confirms that our main results are not significantly influenced by the discontinuity of the SSC sampling method.

The description of this sampling effect has been added to the revised manuscript in the discussion Section 6.3 including the new figure below.

[Figure]

**Figure 2: Cumulative distribution functions of total daily basin–averaged SM (a), IM (b), ER (c) and Q (d), computed on all days and only on days corresponding to SSC–measurements. Only positive values of SM, IM and ER are included in the computations.**

5) Sections 5 & 6: I recommend merging the Results and Discussion. The Results section contains some elements of discussion (e.g. lines 20−25, page 15, on climate patterns), while in multiple locations within the Results section the authors write 'see discussion'. The integration of Results and Discussion could produce a more coherent paper that presents findings and their interpretation in relevant sub−sections. For example, the discussion of snow and icemelt modelling in terms of previously reported melt factors (lines 10−25, page 20) would fit logically with the presentation of the calibration results (section 5.1).

5) We agree that sections of results and discussion can be organized in a more coherent way. We moved the discussion on snowmelt and icemelt models to the results as well as the trend/jump detection analysis.
In the revised manuscript the discussion section now only has two main points:
(1) a discussion of the potential impact of neglecting glacier dynamics, where we demonstrate and discuss why although we are not including glaciers dynamics, we are not significantly overestimating icemelt rates (Sect. 6.1); and
(2) a discussion of other potential anthropogenic factors on our results, such as river channelization, hydropower reservoirs regulation and gravel mining activities, which we argue are not likely to have a significant role in the SSC rise detected in mid−1980s (Sect. 6.2). As possible anthropogenic impact, we consider also the discontinuity of the SSC sampling procedure following the discussion presented at point n.4 and we also discuss the climatic signal on fine sediment dynamics in the context of a heavily regulated catchment.

6) Section 6.2 (lines 15−35, page 21): The discussion of future climate change effects on the sediment regime should be shortened and focus mainly on the point about the value of a more process−based approach. The paper offers no evidence based on future change simulations, so should limit speculative discussion in this area.

6) We agree that the discussion on climate change projections in Switzerland was too extended. However, as suggested by Referee #1, we would like to focus on the importance of adopting a more process−based approach when analysing the effects of climate change on suspended sediments. Therefore, in the revised manuscript we consistently reduced the text but we kept this point present in the discussion in Sect. 6.2.

**Anonymous Referee #2**

In this manuscript the increase in suspended sediments observed in the Rhone river after 1987 (compared to the period 1960−198) is statistically compared to hydro−climatic factors as temperature, rainfall, discharge, snow cover, snow melt and ice melt. Interpolated meteorological products and satellite observation of snow cover were used to model snow cover, snow and ice melt for the Rhone basin with degree−day melt models. The meteorological and the modelled data were then statistically compared to measured discharge and measured sediment supply. From this comparison, the authors suggest that the observed changes in the suspended sediment concentration is mainly due on the one hand to a reduced extent and duration of snow cover leading to larger areas exposed to erosive rainfall and on the other hand to an increase in icemelt in the summer months.

This study represents a valuable contribution for understanding the impact of climatic data on sediment transport. The presented approach is interesting and innovative and the results and analysis are meaningful. The study is also well−documented. Overall the manuscript is well written but substantial improvement can be made on the structure of the paper. This would avoid many repetitions throughout the text. Especially the discussion part needs to be built more on the results of the studies itself to be less speculative (see main comments below). Furthermore, I have some concern on the assumption to not model glacier retreat in the ice model, as this can lead to overestimation of the ice melt. There is therefore a need to further discuss the impact of the simplicity of the ice model on the results (see main comments below). The issues presented in this manuscript are in the scope of HESS and as my comments concern substantial restructuration of the manuscript and further development in the analysis of the ice model results, I would support its publication after major revisions.

We thank Referee #2 for these valuable suggestions. We implemented his/her comments in the revised manuscript as described in the following.

Main comments:
Goal formulation (p. 3, l. 22−25): One of the goals of the study (aim c) is to assess the future climate changes and hydropower operation impacts. This is only discussed later in the discussion, but only in a speculative way as the analysis itself does not include any hydropower data or future climate scenario. As it is formulated in the aims of the study, I would expect future climate scenario being taken into account and hydropower operation being explicitly analyzed. Instead the observations on this impact are only speculative (see for example discussion p. 21, l. 16−33). I would therefore suggest the authors to reformulate their study aims and also to minimize the weight they give to this topic in the discussion as it is only speculative.

We agree with Referee #2. We have reformulated the aims of the study by removing point c (" (c) to discuss the results in the context of future climate change and hydropower operation impacts on sediment budgets."). However, we partially kept the discussion on future climate change investigations to highlight the importance of applying a more process based approach when investigating the impacts of climate change on suspended sediment dynamics (Sect. 6.2).

Overall structure:
The overall structure of the paper leads to many repetitions throughout the text.
1) The main problem is that parts of the discussion can be found in the methods, results, and discussion sections. This makes the text repetitive and the information hard to find. The authors should define a clear structure for their manuscript. They should decide if they want the results and discussion together in one section, meaning a discussion following each results description, or if they want 2 distinct sections (which I personally would recommend), meaning that the result section only describe the results and all discussion and interpretation of the results is moved to discussion section.

We agree with Referee #2 that the structure of the manuscript can be improved. In the revised manuscript, we moved large part of the discussion to the results sections (Sect. 5.1, 5.2, and 5.3) and we discuss separately only two specific aspects (see response to Referee no. 1)

2) The data are described first in the method section and then in the data description section. This is redundant. Best would be to shortly describe the data before the method section.

2) We agree that data description is somewhat redundant. In the revised manuscript we describe all the hydroclimatic variables that we analyse (T, P, Q, SSC, SM, SCF, IM and ER) in Sect. 3 and we list them in Table 1. In Sect. 4 (Data Description) we present all the datasets that we use in this analysis.

3) Section 3.1, snow model:
Several parameters that play an important role in the degree day approach (as SD, Trs, Tsm) are set before calibration. A sensitivity analysis of the impacts of these parameters would be interesting to assess their impact on the results and give them more credit.

3) We agree with Referee #2. We performed a sensitivity analysis on TRS, TSM and $K_{snow}$ as the key parameters of the snow model. We analysed parameters sensitivity in two ways:
(1) we analyze the relative change of some of the goodness of fit measures adopted during the calibration (TSS, NS RMSE) as function of the parameters perturbation, by perturbing one single parameter at the time (Fig. 3);
(2) we estimate the influence of these three parameters on the trends and the jumps that we identified over the period 1975–2015 for the hydroclimatic variables simulated with the model: SM, IM, ER, SCF. For this purpose, we apply statistical tests for equality of mean annual and monthly values before and after mid–1980s after perturbing one parameter at the time within physically reasonable ranges of values.

Results show that the snowmelt factor $k_{snow}$ is the most sensitive parameter (Fig. 3). For $k_{snow}$ between 1.6 and 5.6 mm day$^{-1}$ °C$^{-1}$, the relative reduction of TSS and NS results respectively lower than 10% and 15% (Fig. 3). Although reducing the snowmelt factor $k_{snow}$ below 2.6 mm day$^{-1}$ °C$^{-1}$ increases RMSE by almost 40%, it results in incrementing RMSE only by 0.06 units. For $T_{SM}$ varying within the range of $-2 \div 2$ °C, TSS and NS decrease less than 10%. While the change in RMSE results larger similarly to the case of $k_{snow}$ (Fig. 3). The effect of $T_{RS}$ is even more negligible, with relative changes of goodness of fit measures within 5.5% (Fig. 3).

(a)                                              (b)                                              (c)

[Figure]

**Figure 3: Parameter perturbation analysis on: (a) snowmelt factor $k_{snow}$, (b) threshold temperature for the onset of melt $T_{SM}$, (c) rain–snow threshold temperature $T_{RS}$. The relative change [%] of TSS, NS and RMSE are shown for each parameter.**

In agreement with the results of our analysis (Sect. 5.3), a statistically significant increase of mean annual IM and ER and a statistically significant decrease of mean annual SCF after mid–1980s are detected for all selected parameter values. Similarly, mean annual SM shows a decreasing tendency after mid–1980s for all parameters selections, and in about 50% of the parameters selections statistical tests even show a statistically significant drop in mean annual SM. The increases in mean monthly ER in June and July are identified respectively in more than 90% and in 100% of the cases. Similarly, all parameters sets show an abrupt rise in IM after mid–1980s for all spring and summer months (May–August).

This confirms that by perturbing the parameters of the snow model within reasonable ranges, the overall results of our analysis do not change. This is also depicted in Fig. 4. The differences in mean monthly values of IM and ER between the period after and before mid–1980s are shown for multiple values of each individual parameter. The confidence interval depicted in Fig. 4 is artificially built by selecting, for each month, the highest and the lowest values of the confidence interval (5% significance level) among all parameter sets. By comparing Fig. 4 with Fig. 9c and Fig. 9d of the revised manuscript, it is possible to see that by changing parameters within reasonable ranges would not change significantly the results of our analysis.

These results are presented in the Supplementary Material of the revised manuscript ("S1. Sensitivity Analysis on Snow Model parameters").

[Figure]

**Figure 4: Sensitivity analysis on monthly differences between the periods after and before the year–of–change (1987–2015 and 1975–1986) for IM (left) and ER (right). Box plots represent monthly differences for all selections of parameters within given ranges of values: 1.6 mm day$^{-1}$ °C$^{-1}$ ≤ k$_{snow}$ ≤ 5.6 mm day$^{-1}$ °C$^{-1}$ (a,b); –2°C ≤ T$_{SM}$ ≤ 2 °C (e,f); –1 °C ≤ T$_{RS}$ ≤ 3 °C (c,d). Grey shaded areas represent the widest confidence interval among all selections of the parameters.**

4) Ice model:
In my opinion the most critical point of the method is the simplicity of the ice melt simulation. The ice model uses a simple degree day routine and do not considers changes in glacier areas, which might be consequent over a time period of ca 50 years. The authors claim "Temporal dynamics of glacier coverage are not accounted for —– ice accumulation, glacier retreat and ice movement are disregarded. The reduction of Alpine glaciers for the period 1950–2000 was estimated to be within the 10 range 500– 1000 m (Hoelzle, 2003; Oerlemans, 2005), while our effective climate grid resolution is 2_2 km, i.e. the retreat is considerably lower than the grid resolution of climatic inputs. The consideration of ice dynamics would therefore add a degree of complexity that our spatial resolution cannot take advantage of" (p. 11, l. 8−12). I disagree with this argumentation. The grid size of the meteorological product is surely important for the modelling of glacier retreat, but many studies used these meteorological dataset for modelling changes in glacier mass balance in the Alps and proved that the results were accurate enough. The model resolution for the ice melt calculation is 250 m and is therefore "small" enough to be influenced by a glacier retreat in range of 500−1000 m. My concern is that if the glacier retreat is not taken into account in the modelling of such a long period (50 years, with very high rates of glacier retreat in the last decade), there is a real risk of overestimating the glacier melt by the model. The results of the analysis suggest that there is a shift in the discharge from a snow−melt dominated regime to an icemelt dominated regime (Fig. 10 and 12). I wonder if part of this increase in icemelt can be induced by the fact that glacier retreat is not taken into account. Therefore I would suggest the authors to discuss in

more details the impact of this model assumption on the results. Many glaciers in the Rhone basin are well documented and have yearly mass balance data. It would be worth to compare the modeled ice melt values with measured time series of glacier mass balance to exclude that the increase of ice melt is due to model assumptions.

This is indeed a very important point, and we agree with Referee #2 that neglecting glacier evolution may influence the estimation of icemelt rates. The Referee raises the possibility that by neglecting glacier area loss (retreat) we are possibly overestimating the icemelt contribution over our study period. To provide evidence that this is not the case, we compared our simulations with time series produced with the Global Glacier Evolution Model (GloGEM), a model accounting for both glacier mass balance and evolution. For comparison, we used total monthly runoff (snowmelt + icemelt + rainfall) generated over the glacierized surfaces of the upper Rhone basin, simulated with GloGEM (Huss and Hock, 2015) for the period 1980−2010.

GloGEM computes the mass balance for every glacier in 10−m elevation bands by estimating snow accumulation, snow and ice melt, and refreezing of rain and melt water. The response of glaciers to changes in mass balance is modelled on the basis of an empirical relation between ice thickness change and normalized elevation (Huss et al., 2010). Starting from initial ice thickness values derived by the method of Huss and Farinotti (2012), ice thickness is updated at the end of each hydrological year. GloGEM is calibrated and validated over the period 1980−2010 with estimates of glacier mass changes by Gardner et al. (2013) and in situ measurements provided by the World Glacier Monitoring Service.

Although in our hydrological model – which considers precipitation, snowmelt, icemelt at pixel scale, but integrates them to basin−average values − we do not include glacier evolution, the annual runoff volumes (snowmelt + icemelt + rainfall) from glacierized areas during the period 1980−2010 correlates very well with the results of GloGEM (Fig. 5a). Measures of performance confirm the agreement between the two models: the correlation coefficient is equal to 0.86 and the Nash−Sutcliffe efficiency is equal to 0.67. We are also capable to capture quite well the seasonal pattern of runoff generated from glacierized areas (Fig. 5b).

Perhaps most importantly, GloGEM simulations show that total annual runoff is increasing throughout the period and there is no evidence for decreasing icemelt rates. This confirms that, although glaciers of the upper Rhone basin are retreating, icemelt fluxes originating from glacierized and proglacial areas are increasing during the 1980−2010 period. As expected, total runoff from glacierized surfaces and icemelt are highly correlated (Fig. 5a, correlation coefficient = 0.95), thus indicating that the increase in total runoff is due to an increase of the icemelt component. Indeed, non−parametric Mann−Kendall tests indicate an increasing trend with 5% significant level. Trend slopes, estimated with the Theil−Sen estimator, confirm the agreement between the two models: we find a total runoff of ~27.65 mio m$^3$ year$^{-2}$ with GloGEM and ~21.71 mio m$^3$ year$^{-2}$ with our model. We also computed the basin−averaged mass balance for each hydrological year. The mean mass balance over the period1980−2010 is equal to −0.78 ± 0.22 m w.e./year (Fig 6). This value is slightly greater than that found by Fischer et al. (2015) for the upper Rhone basin (−0.59 m w.e./year), but within the uncertainty of the estimate.

In summary: although we do not account for glaciers retreat, our model results agree well with a state−of the art glaciological model that includes glacier evolution. Both comparisons with GloGEM and our basin−averaged mass balance indicate that we are not significantly overestimating icemelt contribution during the period 1975−2015.

Clearly, if we were looking at future climate projections and at glaciers that are rapidly retreating we would need to include a glacier retreat. Under climate change, even the largest glacier in the basin, the Aletsch Glacier, is expected to shrink at a rate where its icemelt contribution would start decreasing before 2050 (Farinotti et al., 2012; FOEN, 2012; Brönnimann et al., 2014).

We present the results of this analysis in Sect. 6.1 ("Glacier retreat and ice−melt") and we include Fig. 5 in the revised manuscript.

(a)                                                                 (b)

[Figure]

**Figure 5: Runoff (snowmelt + icemelt + rainfall) generated from glacierized areas within the upper Rhone basin, simulated with GloGEM and with our snowmeltsnowmelt and Ice-melt models (degree–day) for the period 1980–2010: (a) total annual values; (b) mean monthly values. Fig. 11a also depicts the time series of total annual Ice-melt simulated with our Ice-melt model.**

[Figure]

**Fig. 6: Mass balance rate for glacierized areas of the upper Rhone basin, simulated with our temperature index model for the period 1975−2015.**

5) Detailed comments:
We thank the reviewer for the following detailed comments. We modified the text accordingly.

p.2, l. 13−18: " In such…": this sentence is too long, make 2 sentences out of it. Isn't rainfall and liquid precipitation the same?
We divided the sentence into two sentences.

p.3, l. 25−27: "Although our results…": move to the discussion.
Done.

p.4, Fig. 1: In my opinion Fig. 1 does not give any additional information. The aim of this figure is not very clear. I would suggest to suppress it or to develop it in a way where it is clear how the 4 sediment sources are build and how they can play a role. Maybe the authors could also link the different sources to the hydro−climatic factors (which factor influences which source).
We agree with the reviewer that Fig. 1 does not give additional information and we removed it.

p.5, l. 1−2: " However … of our investigation": This does not belong to the description of the study site, it belongs rather to the introduction or to the discussion.
We removed this sentence.

p.5, l. 9−11: which are the observed and the modeled data?
We specify in parentheses "observed" and "simulated" to clarify.

p.6, l. 1−3: how do the authors calculate the 250−250m daily temperature and precipitation from the MeteoSwiss gridded dataset (ca. 2x2km)? It is not clear to me if or when you use monthly average over the basin or daily gridded data. Please clarify.
We thank the reviewer for pointing out the necessity of clarifying this point. We changed the text in order: (1) to specify the interpolation method (nearest−neighbor interpolation) applied to obtain the input datasets on a 250×250 m resolution grid; (2) to clarify the time scale of the analysis (daily for the snow and icemelt model simulations, monthly and annual for the analysis of the temporal evolution). We hope it is clearer now.

p.6, Table 1: the gridded dataset from MeteoSwiss is originally in degree; therefore the grid size is ca 2x2km
Indeed the spatial resolution of the gridded dataset is approximately equal to 1.6 km×2.1 km. We agree with the reviewer that is has to be specified by adding the approximation symbol tilde "~".

p.8, l. 4−10: this belongs to the discussion (see also comments above on the ice model). This is also repeated in p.11, l. 8−12.
Regarding Sect. 3.2 (p.8, l. 4−10 of the reviewed manuscript), we changed the text according to the suggestions of the reviewer, while regarding Sect. 4.3 (p.11, l. 8−12 of the reviewed manuscript) we removed the comments because they are now discussed in more details in the discussion Sect. 6.2.

p.9, l. 6−18: Most of this Sect. belongs to the description of the study area, not to the methods section.
We agree and we partly moved this description to the description of the study site (Sect. 2).

p.10, l. 18−21: "we also verified…" And? Where are the results of this analysis? What are the conclusions? I don't understand the sense of these sentences.
We changed this part and we tried to clarify.

p. 10, l. 23−26: this is a repetition.
We reduced the text trying to avoid repetitions.

p.11, l.20: "This value…", it is enough to discuss it only in the discussion, delete this sentence.
We kept this comment in the results since the manuscript has been re−structured.

p.11, l.31−32: belongs to the discussion.
In the revised manuscript, comments on calibration/validation of snow and icemelt models are reported in the results section (5.1) and removed from the discussion section. We hope that this make the text clearer and reduce repetitions.

p.12, l.1: "we are confident" avoid this kind of expression (all over the text) as it gives the impression that the observation are not based on results but on feelings.
We agree with the reviewer and changed the text accordingly.

p.12, l.3−5: "Despite…", it is enough to discuss it only in the discussion, delete this sentence.
In the revised manuscript, comments on calibration/validation of snow and icemelt models are reported in the results section (5.1) and removed from the discussion session. We hope that this make the text clearer and reduce repetitions.

Fig. 3a−b: I am not convinced that these 2 figures bring additional information. I would suggest deleting them.
We removed the figures from the revised manuscript.

p.15, l. 5−8; l.9−11: Move to the discussion.
In the revised manuscript, comments on changes in hydroclimatic variables are reported in the results section (5.2 and 5.3) and removed from the discussion section.

p.15, l.16−24: In this Sect. it is confusing when annual and monthly discharge is meant.
We changed the text in order to differentiate better between mean monthly and mean annual values.

p.15, l.22: "this variability is reflected in streamflow", where?
We deleted this sentence and we keep the discussion on the potential impact of large−scale climatic patterns both on precipitation and discharge.

p.16, l.1−5: move to the discussion.
In the revised manuscript, comments on changes in hydroclimatic variables are reported in the results section (5.2 and 5.3) and removed from the discussion section.

p.16, l.6−10: this part belongs to the discussion and is a repetition.
It was removed from the discussion.

p.16, Fig.8: _Q and _P seems to react inversely, is there an explanation? Is it due to hydropower?
While for discharge hydropower operations might possibly explain the increase in winter (as discussed in the chapter 5.2 of the revised manuscript), we cannot identify a clear connections between precipitation and releases for hydropower reservoirs, on the basis of this dataset. However, we think that this could be an interesting point to investigate more in details in the future. We thank the reviewer for pointing this out.

p.17, l.2: "The fact that sediment supply impacts suspended sediment concentration is evident…" really? Where do I see that?
The sediment rating curve shows that both transport capacity and sediment supply (i.e. sediment sources and sediment production and transport processes) influence suspended sediment concentration. Indeed, given the transport capacity (discharge), suspended sediment concentration covers a wide range of values. We changed the text in order to clarify this point.

p.17, Fig. 9: would it be possible to highlight (different colors) < 1987 and >1987?
We thank the reviewer for this suggestion. We changed the figure by differentiating data for the periods 1965−1986 and 1987−2015.

p.17, l.27: "Our simulation: : :", it is enough to discuss it only in the discussion, delete this sentence.
In the revised manuscript, comments on changes in hydroclimatic variables are reported in the results section (5.2 and 5.3) and removed from the discussion section.

p.17, l.28−30: How did you come to this conclusion? Did you compare it with literature? These sentences belong to the discussion.
We kept this comment in the results since the manuscript has been re−structured. This argument is based on the physical processes occurring in mountainous environments. For example, snow cover influences icemelt both by insulating glacierized surfaces and so preventing melting, and by affecting albedo and therefore the energy balance.

p.18, l.2−6; 8−10: move to the discussion.
We kept this comment in the results since the manuscript has been re−structured.

p.18, l. 16: where is this shown? Cite the figure.
Done.

p.18, l.22: "a shift of snow−melt to ice−melt": I would not call it a shift as it is only slightly visible from fig. 12. Is it statistically significant? Cite the figure where the shift can be seen (fig. 12?).
We agree with the reviewer. In addition, we changed the text in agreement with the figure, which we changed showing the relative contribution of the three hydroclimatic variables during the summer months, when the rise in suspended sediment concentration is observed.

p.19, l.1−15: Most of this belongs to the discussion.
We moved this comment into the result section.

p.20, l. 20−23: this has already been said and is only a repetition.
We removed the entire section 6.1.

p.21, l.10: "…decrease in fine sediment load..." where is this decrease in the 60s to see? The analysis start around 1965, where the sediment load is low, but there is no decrease…
We agree with this comment and we clarified that this decrease was observed and documented by Loizeau et al. (1997) on the basis of sediment cores recovered in the Rhone delta region. We changed the text to clarify this point.

p.21, l. 10−12: how do you come to the conclusion that it can be an effect of hydropower?
This is the hypothesis that has been proposed so far in the literature (Loizeau et al., 1997; Loizeau and Dominik, 2000). The analysis that we conducted in this study does not allow us to reject this hypothesis. However, we demonstrate that climate has a significant effect on suspended sediment dynamics, although the system is highly regulated. Therefore, we argue that together with hydropower operations, the colder temperature that characterizes the 1960s could have contributed to the reduction in suspended sediment concentration observed by previous studies.

p.21, l.16−34: This is only speculative and take too much importance in the discussion.
We agree and we removed it almost completely.

p.22, l.8−10: repetition
We removed it.

p.22, l.14−19: sentence too long, shorten it or separate it.
This sentence was moved to chapter 5.3 and separated into two sentences according to the reviewer.

p.22, l.21: there is no increase in rainfall intensity in the analysis (fig. 7)
Here we refer to the increase in R−factor identified by Meusburger et al. (2012). Although we do not identify any increase in mean annual and mean monthly values of total daily precipitation (Fig. 5b and 6b of the revised manuscript), the analysis of rainfall at sub−daily scale could provide different results. However, we agree that this must be specified and we changed the text accordingly.

p.23, l.18−21: There were 4 sources of sediments in fig. 1, please clarify. To me it is not clear how these sediment sources were taken into account in the analysis. You should maybe emphasis in the discussion how each factor influences each sediment source.
We agree that this needs to be clarified. We changed the term "sediment sources" with the term "sediment fluxes", and we list the three main sediment fluxes that we are analysing, i.e. sediments originated and transported by (1) snowmelt, (2) erosive rainfall, and (3) icemelt. Because we removed Fig 1 and do not specify the sediment sources anymore we think it is not necessary anymore to extend the discussion on how each factor influences each sediment source.

- clarified in the text the tests that we applied on the MeteoSwiss gridded dataset in order to ensure that they are not affected by non-homogeneities (due to the variability of the number of stations used in developing the dataset)
- removed previous Fig. 1 (sediment sources), Fig. 3a ($K_{snow}$) and Fig. 3b ($K_{ice}$);
- changed Fig. 7 (previous Fig. 9) to highlight observations of the period 1965−1986 and 1987−2015. We estimated sediment rating curves separately for the two periods;
- changed Fig. 10 (previous Fig. 12) by depicting the relative contribution of rainfall, snow and icemelt on the sum of these three components, only in July and August, months in which SSC increased.

**Temperature signal in suspended sediment export from an Alpine catchment**

[revised manuscript text omitted]

That said, the construction of dams and start of hydropower operation has coincided with a drop in the suspended sediment load of the main Rhône River measured at la Porte–du–Scex in the 1960s (Loizeau et al., 1997; Loizeau and Dominik, 2000). Two sub–catchments of the Upper Rhône basin are used for the calibration/validation of the icemelt model: the Massa and the Lonza. The Massa (Fig. 1) is a medium–sized basin (195 km$^2$) with a mean elevation of 2 945 m a.s.l. More than 60% of the surface is glacierized, and the remaining surface is classified mostly as rock and firn (Boscarello et al., 2014). The basin includes the Aletsch Glacier, which is the largest glacier in the European Alps with a length of around 23.2 km and a surface area of approximately 86 km$^2$ (Haeberli and Holzhauer, 2003). The Lonza is a relatively small basin located to the west of the Massa (Fig. 1) with an average elevation of 2 630 m a.s.l. It has a total drainage area of 77.8 km$^2$ and its surface consists of 36% of glacier cover. Daily discharge measurements are available for the Massa and for the Lonza respectively at the gauging stations of Blatten bei Naters and Blatten.

[revised manuscript text omitted]
_{\text{ice}}$ with data from two highly glacierized sub–basins in the Rhone: Massa and Lonza (Fig. 1). The calibration method is described in Sect. 3.3.

**3.3 Calibration and Validation**

We perform the calibration and validation of the snow and icemelt model parameters in sequence, since the snow–covered surface is required for icemelt estimation on glaciers. The snowmelt factor $k_{\text{snow}}$ is calibrated based on comparisons with snow cover data derived from satellite images (MODIS). Snow cover observations are split into two periods: 1 October 2000 – 30 September 2005 for calibration and 1 October 2005 – 31 December 2008 for validation. MODIS maps of snow cover are filtered to reduce the impacts of clouds on SCF estimations. The resulting number of calibration and validation days is equal to 217 and 143 respectively. Snow cover maps at 500×500 m resolution are distributed by proximal interpolation to the snowmelt model 250×250 m computational grid. Snow depth maps simulated with Eq. (4) and transformed into a simulated snow cover fraction $SCF^{\text{sim}}$ in Eq. (5) are compared with the MODIS maps $SCF^{\text{obs}}$.

The objective function for calibration is based on a combination of mean absolute error and true skill statistic. The mean absolute error MAE is estimated as:

$$MAE = \frac{1}{n}\sum_{j=1}^{n}\left|SCF_j^{\text{obs}} - SCF_j^{\text{sim}}\right| \; , \tag{9}$$

where n is the number of MODIS image maps, and it captures the overall ability of the model to reproduce the snow cover fraction accurately. The true skill statistic TSS is a spatial statistic that measures the grid–to–grid performance of the model in capturing snow–no snow presence. It is computed as the sum of sensitivity SE (correct snow predictions) and specificity SP (correct no–snow predictions) computed from contingency tables (e.g. Wilks, 1995; Mason and Graham, 1999; Corbari, 2009) in each image j and averaged over the n MODIS maps in the simulation period:

$$TSS = \frac{1}{n}\sum_{j=1}^{n} TSS_j = \frac{1}{n}\sum_{j=1}^{n} SE_j + SP_j - 1 \tag{10}$$

Because TSS includes both sensitivity and specificity, it captures both predictions of snow–covered and snow–free areas. It takes on values between 0 and 1, where 1 indicates perfect performance. TSS is a widely applied metric for assessing spatial model performance (e.g. Begueria, 2006; Allouche et al., 2006). We combine both goodness–of–fit measures (MAE and TSS) into an objective function OF, by giving more weight to MAE. Finally, we evaluate the objective function OF over $k = 5$ different elevation bands in order to better capture the topographic gradients in snowmelt distribution in the Rhône Basin:

$$OF = \sum_{k=1}^{5} OF_k = \sum_{k=1}^{5} -0.6 \, MAE_k + 0.4 \, TSS_k \; . \tag{11}$$

This objective function is maximized in calibration. The rationale of using both MAE and TSS in evaluating performance is to give weight to both basin–integrated snow cover as well as to grid–based predictions. Indeed, the same value of snow cover fraction can result in two different spatial arrangements of snow–covered pixels, and a correct spatial distribution of snow–covered and snow–free areas is relevant for this analysis insofar as it affects the activation and deactivation of specific sediment sources. The weights assigned to MAE and TSS in Eq. (11) are the outcome of sensitivity tests with the model. After calibration, we also estimate the Nash–Sutcliffe efficiency NS (Nash and Sutcliffe, 1970) and the mean square error MSE to quantify the performance of the model:

$$NS = 1 - \frac{\sum_{j=1}^{n}(SCF_j^{\text{obs}} - SCF_j^{\text{sim}})^2}{\sum_{j=1}^{n}(SCF_j^{\text{obs}} - \overline{SCF})^2} \; , \tag{12}$$

[revised manuscript text omitted]

**5.3 Hydroclimatic Activation of Sediment Sources**

The concentration of suspended sediment transported at the outlet of the catchment does depend both on the transport capacity (discharge) and on the sediment supply. Indeed, given the same discharge (transport capacity), actual suspended sediment concentration covers a wide range of values (Fig. 7). Sediment supply depends on many factors, most importantly the spatial location of sediment sources (e.g. different lithology, distance to outlet, connectivity) and the specific processes of sediment production (e.g. hillslope erosion, glacial erosion, release of subglacially stored sediment, channel bed and bank erosion, mass wasting events) and transport (e.g., hysteresis). All these factors contribute to the variability around sediment rating curves (Fig. 7). Since we demonstrate that discharge has not changed significantly, the increase in suspended sediment concentration observed at the outlet of the Rhône Basin during mid–1980s is more likely related to altered sediment supply

conditions than to a greater transport capacity. Therefore, to investigate the link between the rise in mean air temperature and suspended sediment concentration, we consider here the three main sediment production and transfer processes typical of Alpine environments: (1) the continuous effect of snowmelt runoff on hillslope and channel erosion, (2) the intermittent but potentially considerable contribution of hillslope, channel bed and bank erosion, and mass wasting triggered by rainfall events, (3) the sediment–rich flux coming from glacierized areas during the icemelt season. We analyse the time series of simulated snowmelt, snow cover fraction, icemelt and effective rainfall at annual (Fig. 8) and at monthly (Fig. 9) timescales, to identify possible changes in those variables around 1987. Results are described in the following.

[revised manuscript text omitted]
 (snowmelt + icemelt + rainfall) from glacierized areas during the period 1980−2010 correlates very well with the results of GloGEM (Fig. 11a). Measures of performance confirm the agreement between the two models: the correlation coefficient is equal to 0.86 and the Nash−Sutcliffe efficiency is equal to 0.67. We are also capable to capture quite well the seasonal pattern of runoff generated from glacierized areas (Fig. 11b). Perhaps most importantly, GloGEM simulations show that total annual runoff is increasing throughout the period and there is no evidence for decreasing icemelt rates. This confirms that, although glaciers of the upper Rhône basin are retreating, melt−water discharges from glacierized and proglacial areas are increasing during the 1980−2010 period. As expected, total runoff from glacierized surfaces and icemelt are highly correlated (Fig. 11a, correlation coefficient = 0.95), thus indicating that the increase in total runoff is due to an increase of the icemelt component. Indeed, non−parametric Mann−Kendall tests indicate an increasing trend with 5% significant level. Trend slopes, estimated with the Theil−Sen estimator, confirm the agreement between the two models: we find a total runoff of ~27.65 mio m$^3$ year$^{-2}$ with GloGEM and ~21.71 mio m$^3$ year$^{-2}$ with our model. We also computed the basin−averaged mass balance accounting for snow accumulation and snow and icemelt for each hydrological year. The mean mass balance over the period 1980−2010 is equal to −0.78 ± 0.22 m w.e. year$^{-1}$ which is within the uncertainty range of recent studies (Fischer et al., 2015). In summary: although we do not account for glaciers retreat, our model results agree well with a state−of the art glaciological model that includes glacier evolution. Both comparisons with GloGEM and our basin−averaged mass balance indicate that we are not significantly overestimating icemelt contribution during the period 1975−2015. We also compute the basin–averaged mass balance accounting for snow accumulation and snow and icemelt for each hydrological year. The mean mass balance rate over the period 1980–2010 is equal to –0.78±0.22 m w.e. year$^{-1}$ which is within the uncertainty range of recent studies (Fischer et al., 2015). We therefore argue that although we do not account for glacier retreat, our model results agree with with a state−of the art glaciological model that includes glacier evolution. Both comparisons with GloGEM and our basin– averaged mass balance indicate that we are unlikely to be overestimating the icemelt contribution during the period 1975– 2015. Clearly, if we were looking at future climate projections and at glaciers that are rapidly retreating we would need to include a glacier retreat. Under climate change, even the largest glacier in the basin, the Aletsch Glacier, is expected to

[revised manuscript text omitted]

---

## Referee Report (RR1)

Review for the manuscript « Temperature signal in suspended sediment export from an Alpine catchment » by Costa et al.

In the presented study, the increase in suspended sediments in the Rhone river after 1987 is statistically analysed and related to hydroclimatic factors and the authors put much efforts into improving their manuscript. The authors improved their manuscript including a more in-depth analysis of the results and more explanations on the methodology and its limitation (sensitivity analysis, discussion on the ice model, …), which highly increased the quality of the manuscript. Even if the structure in this new version is clearly improved, it still has some issues, that unfortunately still make the study difficult to follow. My concern is that lot of information is spread around the text making the text often repetitive and difficult to follow, since it is often difficult to know where the information can be found. I list some advices below in order to improve the structure of the manuscript. I would recommend the manuscript for publication after some structural revisions.

I disagree with the authors to put part of the discussion in the results and part of the discussion in the discussion part. It makes on one hand the main message of the manuscript difficult to follow and on the other hand it gives to much wait on the ice melt model and on the anthopogenic effects, which both are not the main part of the analysis. This is confusing for the reader and the main meassage get lost. I would recommend to clearly distinguish the result part and the discussion part. and then to first briefly explain the ice model (results (figure 11 and 12 should be presented in the result part) and then to give the discussion on the main topics, in order to give it more weight.

The introduction contains a large amount of information but it is not straight forward to me what are the main points and goals listed there. The paragraphs 2 and 3 (p.2 l.15-34) are a good example for my concern as I find the entire paragraphs difficult to follow and especially some information is only half given. For example on line 19 : « --- the seasonal dynamics of ice-melt, also directly affect sediment supply. » How does it affect sediment supply ? this is first later explained, partly in the introduction and partly in the methods (p. 4 l. 18-25). I suggest to move everything to the introduction. It makes sense to clearly explain the importance of the different variables in the introduction, to directly make the choice of the variables clear to the reader.

The introduction also contains several repetitions as for example on p.2, l.37-38 « Alpine regions represent… » that was already said on the same page line 1-3.

The two last paragraphs present the reasearch gaps and the goals of the study. Even if they are better formulated than in the last version, it was still not complitely clear to me what was the solution the authors suggested to test in their study since traditional rating-curve analysis does not work in their case (p.3, l.4-9). I find the objectives described at the begining of the methods (p.4, l.14-25) clearer than in the introduction. I would remove them from the method part (where only the method should be described) and include it to the introduction.

p.6, l.15-16 : « MODIS maps of snow cover are filtered to reduce the impacts of clouds on SCF estimation. » How are they filtered ? This is first explained later in the manuscript.

p.15, l..10-14 : « Altough... » move this to the conclusion.

Check thourghout the text : snow-free and snow free (decide which one to use)

p.2, l. 41 : avoid the word « much »

p.3, l.2 : « ...at least ...» why at least ? what would be the other variables ? And why are they less important ?

p.4, l.1-2 : already in introduction

p.4, l.9 : from which year is the glacier area ?

p.4, l.12 : for which period do you have discharge measurement at the Massa ?

p.4, l.26 : which variables are from observations or from interpolation ?

p.4, l.35 : Has SCF been defined before ? Please reformulate the entire sentence.

Variable names : There is overall lots of variable names. Are SM and M_snow really different variable or SM be M_snow_mean ? Same for ice. It would make the variable names easier to follow.

p.5, l.30 : T_sm=0°C, is it the same as in litereature ?

Equation 11 : k was already used for melt factor, use another name.

p.8, l.4 : « Results confirm the validity of the original datasets... » Which results ? how do they confirm this ? References ?

p.8, l.18 : what is the resolution of the snow cover maps ?

In the manuscript : The abbrevations GLIMS and FOEN are not explained.

p.8, l.25 : Earlier it was said that glacier covers come from satellite. Why now from GLIMS ? please explain.

Overall : avoid the word very.

p.9, l.33 : with upper, is « over » meant ?

p.9,l.39-40 : I would not use « we recommand » as it is not the result of your analysis.

p.9, l.30 : « In Figure 4,... » reformulate the sentence.

p.10, l.10-15 : mg l-1 (-1 should be upperscript), plus the sentences in this paragraph are too long.

p.10, l.18 : where is this significant change to see ? give the figure number.

p.10, l.26 : these OSCILLATIONS may be caused...

p.10,l.35 : doesn't is spoken language, replace with does not.

p.10, l.40 : delete « (transport capacity) » as it is already said and is confusing.

p.11, l.1-11 : belongs to the methods

p.11, l.13 : « due to poor snow co ver » refer to fig.9b

p.11, l33-36 : belongs to methods.

p.12 , l.12 : …sediment concentration in those months » refer to figure 6c.

p.12, l.16 : « more than 30% » where is this to see ?

p12, l.40 -43 : why is glacier melt the main cause and not ER ? PLease be more precise.

p.13, l.14-22 : move to methods.

p.14, l.5-9 : end on a positive note otherwise it disqualify your method.

P14, l.30 replace by by be.

Figure 9 : (d) should be delta ER (instead of IM) on the y-axis. In the description : mean change in (a) snow  melt, (b) snow cover fraction,…

---

## Author Response (AR2)

**Reply to review [paper hess-2017-2]**

**Temperature signal in suspended sediment export from an Alpine catchment**

Anna Costa, Peter Molnar, Laura Stutenbecker, Maarten Bakker, Tiago A. Silva, Fritz Schlunegger, Stuart N. Lane, Jean–Luc Loizeau, Stéphanie Girardclos

We thank the Editor and the reviewer for their comments. We have analysed their suggestions and we have revised the manuscript accordingly. We report in the following our response.

**Editor**

*Dear authors,*
*following up on the assessment of the revised version of your manuscript, a few (rather minor) issues remain to be addressed before acceptance for publication. There have been substantial improvements made - but a relatively important concern still lies in the structuring of the paper. I agree with referee 2 that your contribution would largely benefit from a careful separation between elements related to results and those belonging to the discussion. It is not always straightforward to make that clear distinction, but it is nonetheless an important point that greatly helps the reader to find her/his way through the manuscript. A few additional other points listed by referee 2 might equally be addressed. Overall, the required changes should not be too difficult to implement before further processing of your manuscript.*

We thank the Editor for his comment. We have re-structured the manuscript accordingly. We have separated results and discussion and we have addressed the suggestions of the Referee #2.

**Referee #2**

*In the presented study, the increase in suspended sediments in the Rhone River after 1987 is statistically analysed and related to hydroclimatic factors and the authors put much efforts into improving their manuscript. The authors improved their manuscript including a more in-depth analysis of the results and more explanations on the methodology and its limitation (sensitivity analysis, discussion on the ice model, ...), which highly increased the quality of the manuscript. Even if the structure in this new version is clearly improved, it still has some issues, that unfortunately still make the study difficult to follow. My concern is that lot of information is spread around the text making the text often repetitive and difficult to follow, since it is often difficult to know where the information can be found. I list some advices below in order to improve the structure of the manuscript. I would recommend the manuscript for publication after some structural revisions.*

We thank Referee #2 for her/his comment. We have revised the manuscript according to her/his suggestions and we report in the following our response.
* * *
1) *I disagree with the authors to put part of the discussion in the results and part of the discussion in the discussion part. It makes on one hand the main message of the manuscript difficult to follow and on the other hand it gives to much wait on the ice melt model and on the anthropogenic effects, which both are not the main part of the analysis. This is confusing for the reader and the main message get lost. I would recommend to clearly distinguish the result part and the*

*discussion part. And then to first briefly explain the ice model results (figure 11 and 12 should be presented in the result part) and then to give the discussion on the main topics, in order to give it more weight.*

We agree with Referee #2 that a clear distinction between results and discussion may improve the readability of the manuscript and bring into focus the main message of this work. We have re-structured the paper according to this suggestion.

In the revised manuscript, results are organized in 4 sections:
- 5.1 Calibration of Snowmelt and Icemelt Models;
- 5.2 Temperature, Precipitation, Discharge and SSC in the Rhône Basin;
- 5.3 Hydroclimatic Activation of Sediment Sources;
- 5.4 Effect of Intermittent SSC sampling.

The discussion section contains the discussion of the first three result sections:
- 6.1 Snowmelt and Icemelt Models;
- 6.2 Changes in Hydroclimatolology and SSC
- 6.3 Hydroclimatic Activation of Sediment Sources.

The potential impact of anthropogenic factors (dams, channelization and gravel mining) and climate on the sediment dynamics of the catchment is discussed in a separate section (6.4 Anthropogenic Factors and Climate Signals).

As suggested by the Reviewer, we moved the results of the comparison between our icemelt model and GloGEM (previous Fig. 12, current Fig. 5) and the results of the potential effect of the intermittent SSC sampling (previous Fig. 12 and current Fig. 11) into the result section, respectively in Sect. 5.1 and Sect. 5.4.
* * *
*2) The introduction contains a large amount of information but it is not straight forward to me what are the main points and goals listed there. The paragraphs 2 and 3 (p.2 l.15-34) are a good example for my concern as I find the entire paragraphs difficult to follow and especially some information is only half given. For example on line 19: « ...the seasonal dynamics of icemelt, also directly affect sediment supply.» How does it affect sediment supply? This is first later explained, partly in the introduction and partly in the methods (p. 4 l. 18-25). I suggest to move everything to the introduction. It makes sense to clearly explain the importance of the different variables in the introduction, to directly make the choice of the variables clear to the reader.*

*The introduction also contains several repetitions as for example on p.2, l.37-38 « Alpine regions represent… » that was already said on the same page line 1-3.*

*The two last paragraphs present the research gaps and the goals of the study. Even if they are better formulated than in the last version, it was still not completely clear to me what was the solution the authors suggested to test in their study since traditional rating-curve analysis does not work in their case (p.3, l.4-9). I find the objectives described at the beginning of the methods (p.4, l.14-25) clearer than in the introduction. I would remove them from the method part (where only the method should be described) and include it to the introduction.*

We agree with Referee #2 that the introduction section could benefit from a re-organization of the information. We partially removed the second paragraph (pg. 2, l. 14-19), we deleted the discussion on sediment rating curves (pg. 3, l. 3-9) and the sentence on pg. 2 l. 37-38. In addition, we moved the description of the hydroclimatic variables that we analyse from the methods (Sect. 3) to the introduction section (pg. 3, l. 9-16). We also changed the order of the chapters to make the reading clearer and more straight forward.
* * *
*3) p.6, l.15-16 : « MODIS maps of snow cover are filtered to reduce the impacts of clouds on SCF estimation. » How are they filtered? This is first explained later in the manuscript.*

We removed this comment from this section.

*4)  p.15, l.10-14: «Although… » move this to the conclusion.*
We moved this comment to the conclusion section (pg. 15, l. 29–33 of the revised manuscript).

*5)  Check throughout the text : snow-free and snow free (decide which one to use)*
Done.

*6)  p.2, l. 41: avoid the word « much »*
Done.

*7)  p.3, l.2: « ...at least ...» why at least? what would be the other variables ? And why are they less important?*
We changed this sentence (pg. 2, l. 15–17 of the revised manuscript).

*8)  p.4, l.1-2 : already in introduction*
Removed.

*9)  p.4, l.9: from which year is the glacier area?*
In the revised manuscript (pg. 4, l. 3–7), we report the reference year for Aletsch Glacier length and extension (1973) and for the glacier coverage of the sub-catchment Lonza (1991).

*10) p.4, l.12: for which period do you have discharge measurement at the Massa?*
Daily discharge is available at Blatten bei Naters (Massa) since 1931 and at Blatten (Lonza) since 1956. We removed this sentence from the Sect. 2 and we added this information in Sect. 4.2 (pg. 8, l. 17–20 of the revised manuscript).

*11) p.4, l.26: which variables are from observations or from interpolation?*
In the revised manuscript we specified: Q and SSC datasets come from observations, while P and T datasets come from interpolation of observations (pg. 4, l. 12–14 of the revised manuscript).

*12) p.4, l.35: Has SCF been defined before ? Please reformulate the entire sentence.*
We have specified SCF in the introduction and in Sect. 3 (pg. 3, l. 11 and pg. 4, l. 12 of the revised manuscript), therefore we consider no longer necessary to change this sentence.

*13) Variable names: There is overall lots of variable names. Are SM and M_snow really different variable or SM be M_snow_mean ? Same for ice. It would make the variable names easier to follow.*
We agree with Referee #2 that different names for the spatially distributed and the basin–averaged snow and icemelt rates can create confusion. However, we prefer to keep the simplest names for the basin–averaged values because they are mentioned more frequently in the manuscript. Therefore, we have renamed the snow and icemelt rates of individual cells i from $M_{snow_i}$ and $M_{ice_i}$ to $SM_i$ and $IM_i$ and we have kept the basin–averaged names as SM and IM (see Sect. 3.1 and 3.2 of the revised manuscript).

*14) p.5, l.30: T_sm=0°C, is  it the same as in literature ?*
$T_{SM} = 0°C$ is often chosen as temperature threshold for snowmelt in Alpine regions. We changed the text and we added references to three previous works that used this value (pg. 5, l. 20–21 of the revised manuscript).

*15) Equation 11: k was already used for melt factor, use another name.*
Changed to b (pg. 6, l. 22 of the revised manuscript).

*16) p.8, l.4: « Results confirm the validity of the original datasets… » Which results? how do they confirm this? References?*
We agree that this sentence is not clear and we changed it into: "…We applied the statistical tests for detecting changes (Sect. 3.4) both on the original and the experimental datasets of P and T. Results of the statistical tests on the two datasets

coincide. This confirms that temporally variable number of meteorological stations employed to build the product does not influence the changes detected in the original dataset..." (pg. 8, l. 12–15 of the revised manuscript).

*17) p.8, l.18: what is the resolution of the snow cover maps?*
We indicate the resolution of the snow cover maps two sentences after (pg. 8, l. 31 of the revised manuscript).

*18) In the manuscript: The abbreviations GLIMS and FOEN are not explained.*
We thank the Referee for pointing this out. We added the definitions (pg. 8, l. 17 and l. 35 of the revised manuscript).

*19) p.8, l.25: Earlier it was said that glacier covers come from satellite. Why now from GLIMS? please explain.*
Because the GLIMS project uses also historical information derived from maps and aerial photographs, in addition to satellite data, we agree with Referee #2 that this could create confusion. Therefore, in Sect. 3.2 we removed "… from remote sensing data …".

*20) Overall: avoid the word very.*
Done.

*21) p.9, l.33: with upper, is « over » meant?*
We removed this sentence from the revised manuscript.

*22) p.9, l.39-40: I would not use « we recommend » as it is not the result of your analysis.*
We removed this sentence from the revised manuscript.

*23) p.9, l.30: « In Figure 4, … » reformulate the sentence.*
We have reformulated the sentence (pg. 9, l. 23–25 of the revised manuscript).

*24) p.10, l.10-15: mg l-1 (-1 should be upperscript), plus the sentences in this paragraph are too long.*
We corrected the unit measures and we changed the text to reduce the length of the sentences (pg. 10, l. 11–13 of the revised manuscript).

*25) p.10, l.18: where is this significant change to see? give the figure number.*
To clarify this point we changed this sentence into: "… Suspended sediment concentration is also characterized by much larger inter–annual variability after 1987 than before: the standard deviation of mean annual SSC increases from ~ 32 mg $l^{-1}$ before 1987 up to ~ 78 mg $l^{-1}$ after (Fig. 6c). A statistically significant increase in the variance is confirmed with a two–sample F–test at 5% significant level…" (pg. 10, l. 14–17 of the revised manuscript).

*26) p.10, l.26: these OSCILLATIONS may be caused…*
We removed this sentence from the revised manuscript.

*27) p.10, l.35: doesn't is spoken language, replace with does not.*
Done.

*28) p.10, l.40: delete « (transport capacity) » as it is already said and is confusing.*
Done.

*29) p.11, l.1-11: belongs to the methods.*
We moved this paragraph to the method.

*30) p.11, l.13: « due to poor snow cover » refer to fig.9b.*
Done.

*31) p.11, l33-36: belongs to methods.*
Done.

*32) p.12, l.12: ...sediment concentration in those months » refer to figure 6c.*
Done.

*33) p.12, l.16 « more than 30% » where is this to see?*
In the revised manuscript, we indicate the relative contributions of SM and IM before after 1987 and we add reference to the relative figure (Fig. 10) (pg. 11, l. 3–6 of the revised manuscript).

*34) p12, l.40 -43: why is glacier melt the main cause and not ER? Please be more precise.*
We added the following sentence (pg. 13, l. 38–39 of the revised manuscript) to clarify: "…As shown in Sect. 5.3, icemelt increase is highest in July and August (Fig. 9c), in agreement with the jump in suspended sediment concentration (Fig. 7c), while ER rise occurred mainly in June and July (Fig. 9d). We then conclude that ...".

*35) p.13, l.14-22: move to methods.*
Done.

*36) p.14, l.5-9: end on a positive note otherwise it disqualify your method.*
We changed this sentence into: "… However, considering climate projections further into the future, and glaciers that continue to retreat, the issue of future icemelt contribution will need to be revised. Under climate change, even the largest glacier in the basin, the Aletsch Glacier, is expected to shrink at a rate where its icemelt contribution would start decreasing before 2050 (Farinotti et al., 2012; FOEN, 2012; Brönnimann et al., 2014)…" (pg. 12, l. 8–11).

*37) P14, l.30 replace by by be.*
Done.

*38) Figure 9: (d) should be delta ER (instead of IM) on the y-axis. In the description: mean change in (a) snow melt, (b) snow cover fraction,…*
We thank the Referee for pointing this out. We changed the label of the y-axis of previous Fig. 9d (current Fig. 8d) and we modified the caption of the figure.

**List of all relevant changes**

We re-structured the manuscript by separating more clearly results (Sect. 5) and discussion (Sect. 6) and by moving some content and related figures of the previous discussion section to the new result section.

We re-organized and reduced the introduction (Sect. 1) and we changed the reference list accordingly. We moved the description of the analyzed hydroclimatic variables from the methods (Sect. 3) to the introduction section (Sect. 1).

We removed the discussion on rating curve uncertainty and previous Fig. 7. We changed the numbers of the figures accordingly.

We corrected the label of the y-axis of Fig. 9 from IM to ER.

We changed the names of the variables representing the melting rates of individual cells from $M_{ice}$ to $IM_i$ and from $M_{snow}$ to $SM_i$.

We changed the text to increase the clarity of the manuscript according to the detailed comments of Referee #2.

In the followings, is a marked-up version of the previous manuscript with highlighted the main changes. In grey color are highlighted the parts deleted from the previous version, and in light blue color are highlighted the parts added in the revised manuscript.

**Temperature signal in suspended sediment export from an Alpine catchment**

[revised manuscript text omitted]

That said, the construction of dams and start of hydropower operation has coincided with a drop in the suspended sediment load of the main Rhône River measured at la Porte–du–Scex in the 1960s (Loizeau et al., 1997; Loizeau and Dominik, 2000). Two sub–catchments of the Upper Rhône basin are used for the calibration/validation of the icemelt model: the Massa and the Lonza. The Massa (Fig. 1) is a medium–sized basin (195 km$^2$) with a mean elevation of 2 945 m a.s.l. More than 60% of the surface is glacierized, and the remaining surface is classified mostly as rock and firn (Boscarello et al., 2014). The basin includes the Aletsch Glacier, which is the largest glacier in the European Alps with a length of around 23.2 km and a surface area of approximately 86 km$^2$ (Haeberli and Holzhauer, 2003). The Lonza is a relatively small basin located to the west of the Massa (Fig. 1) with an average elevation of 2 630 m a.s.l. It has a total drainage area of 77.8 km$^2$ and its surface consists of 36% of glacier cover. Daily discharge measurements are available for the Massa and for the Lonza respectively at the gauging stations of Blatten bei Naters and Blatten.

[revised manuscript text omitted]
_{\text{ice}} = k_{\text{ice}}(T_{\text{mean}} - T_{\text{IM}}) & T_{\text{mean}} > T_{\text{IM}} \\ M_{\text{ice}} = 0 & T_{\text{mean}} \leq T_{\text{IM}} \end{cases},\tag{7}$$

$$\begin{cases} \text{IM}_i = k_{\text{ice}}(T_{\text{mean}} - T_{\text{IM}}) & T_{\text{mean}} > T_{\text{IM}} \\ \text{IM}_i = 0 & T_{\text{mean}} \leq T_{\text{IM}} \end{cases},\tag{7}$$

where $T_{\text{mean}}$ [°C] is mean daily air temperature, $T_{\text{IM}}$ [°C] is a threshold temperature for the onset of ice–melt, and $k_{\text{ice}}$ [mm day$^{-1}$ °C$^{-1}$] is the icemelt factor. For the entire catchment, we estimate mean daily icemelt IM [mm day$^{-1}$] as the arithmetic average over all ice–covered grid cells:

$$\text{IM(t)} = \frac{1}{N}\sum_{i=1}^{N} M_{\text{ice}\,i}(t)\, \text{IM}_i(t) .\tag{8}$$

The threshold temperature for glacier melting $T_{\text{IM}}$ is set equal to 0 °C. Icemelt occurs only if the glacier cell is snow–free. The snow cover simulated by the snowmelt model in Sect. 3.1 is thus essential for estimating ice–melt. The calibration of the icemelt model consists of estimating the melt factor $k_{\text{ice}}$ with data from two highly glacierized sub–basins in the Rhone: Massa and Lonza (Fig. 1). The calibration method is described in Sect. 3.3.

**3.3 Calibration and Validation**

[revised manuscript text omitted]
 icemelt we recommend to use approaches based on the energy balance (e.g. Pellicciotti et al., 2005).

Although in our hydrological model we do not include glacier evolution, the annual runoff volumes (SM+IM+R) from glacierized areas during the period 1980−2010 correlates well with the results of GloGEM (Fig. 5a). Measures of performance confirm the agreement between the two models: the correlation coefficient is equal to 0.86 and the Nash−Sutcliffe efficiency is equal to 0.67. We also capture quite well the seasonal pattern of runoff generated from glacierized areas (Fig. 5b). Perhaps most importantly, GloGEM simulations show that total annual runoff is increasing throughout the period and there is no evidence for decreasing icemelt rates. This confirms that, although glaciers of the upper Rhône basin are retreating, melt−water discharges from glacierized and proglacial areas are increasing during the 1980−2010 period. As expected, total runoff from glacierized surfaces and icemelt is highly correlated (Fig. 5a, correlation coefficient = 0.95), thus indicating that the increase in total runoff is due to an increase of the icemelt component. Indeed, non−parametric Mann−Kendall tests indicate an increasing trend with 5% significant level. Trend slopes, estimated with the Theil−Sen estimator, confirm the agreement between the two models: we find a total runoff of ~27.65 mio m$^3$ year$^{-2}$ with GloGEM and ~21.71 mio m$^3$ year$^{-2}$ with our model. We also computed the basin−averaged mass balance accounting for snow accumulation and snow and icemelt for each hydrological year. The mean mass balance over the period 1980−2010 is equal to −0.78 ± 0.22 m w.e. year$^{-1}$ which is within the uncertainty range of recent studies (Fischer et al., 2015). In summary: although we do not account for glacier retreat, our model results agree well with state−of the art glaciological models that

include glacier evolution. Both comparisons with GloGEM and our basin−averaged mass balance indicate that we are not significantly overestimating icemelt during the period 1975−2015.

**5.2 Temperature, Precipitation, and SSC in the Rhône Basin, Discharge and SSC in the Rhône Basin**

[revised manuscript text omitted]

**5.3 Hydroclimatic Activation of Sediment Sources**

The concentration of suspended sediment transported at the outlet of the catchment does depend both on the transport capacity (discharge) and on the sediment supply. Indeed, given the same discharge (transport capacity), actual suspended sediment concentration covers a wide range of values (Fig. 7). Sediment supply depends on many factors, most importantly the spatial location of sediment sources (e.g. different lithology, distance to outlet, connectivity) and the specific processes of sediment production (e.g. hillslope erosion, glacial erosion, release of subglacially stored sediment, channel bed and bank erosion, mass wasting events) and transport (e.g., hysteresis). All these factors contribute to the variability around sediment rating curves (Fig. 7). Since we demonstrate that discharge has not changed significantly, the increase in suspended sediment concentration observed at the outlet of the Rhône Basin during mid–1980s is more likely related to altered sediment supply conditions than to a greater transport capacity. Therefore, to investigate the link between the rise in mean air temperature and suspended sediment concentration, we consider here the three main sediment production and transfer processes typical of Alpine environments: (1) the continuous effect of snowmelt runoff on hillslope and channel erosion, (2) the intermittent but potentially considerable contribution of hillslope, channel bed and bank erosion, and mass wasting triggered by rainfall events, (3) the sediment–rich flux coming from glacierized areas during the icemelt season. We analyse the time series of simulated snowmelt, snow cover fraction, icemelt and effective rainfall at annual (Fig. 8) and at monthly (Fig. 9) timescales, to identify possible changes in those variables around 1987. Results are described in the following.

[revised manuscript text omitted]
 (snowmelt + icemelt + rainfall) from glacierized areas during the period 1980−2010 correlates very well with the results of GloGEM (Fig. 11a). Measures of performance confirm the agreement between the two models: the correlation coefficient is equal to 0.86 and the Nash−Sutcliffe efficiency is equal to 0.67. We are also capable to capture quite well the seasonal pattern of runoff generated from glacierized areas (Fig. 11b). Perhaps most importantly, GloGEM simulations show that total annual runoff is increasing throughout the period and there is no evidence for decreasing icemelt rates. This confirms that, although glaciers of the upper Rhône basin are retreating, melt−water discharges from glacierized and proglacial areas are increasing during the 1980−2010 period. As expected, total runoff from glacierized surfaces and icemelt are highly correlated (Fig. 11a, correlation coefficient = 0.95), thus indicating that the increase in total runoff is due to an increase of the icemelt component. Indeed, non−parametric Mann−Kendall tests indicate an increasing trend with 5% significant level. Trend slopes, estimated with the Theil−Sen estimator, confirm the agreement between the two models: we find a total runoff of ~27.65 mio m$^3$ year$^{-2}$ with GloGEM and ~21.71 mio m$^3$ year$^{-2}$ with our model. We also computed the basin−averaged mass balance accounting for snow accumulation and snow and icemelt for each hydrological year. The mean mass balance over the period 1980−2010 is equal to −0.78 ± 0.22 m w.e. year$^{-1}$ which is within the uncertainty range of recent studies (Fischer et al., 2015). In summary: although we do not account for glaciers retreat, our model results agree well with a state−of the art glaciological model that includes glacier evolution. Both comparisons with GloGEM and our basin−averaged mass balance indicate that we are not significantly overestimating icemelt contribution during the period 1975−2015.   We also compute the basin–averaged mass balance accounting for snow accumulation and snow and icemelt for each hydrological year. The mean mass balance rate over the period 1980–2010 is equal to –0.78±0.22 m w.e. year$^{-1}$ which is within the uncertainty range of recent studies (Fischer et al., 2015). We therefore argue that although we do not account for glacier retreat, our model results agree with with a state−of the art glaciological model that includes glacier evolution. Both comparisons with GloGEM and our basin–averaged mass balance indicate that we are unlikely to be overestimating the icemelt contribution during the period 1975–2015. Clearly, if we were looking at future climate projections and at glaciers that are rapidly retreating we would need to include a glacier retreat. 
[revised manuscript text omitted]

Finally, a possible anthropogenic effect on the results may come for the intermittent sampling of SSC (twice per week) which may affect the changes found in the observation period. We estimate this potential effect by considering total daily, basin–averaged values of the hydroclimatic variables SM, IM, ER and Q only on days corresponding to SSC–measurement days. We compare these new time series ("SSC–measurement days") with the original ones ("all days") by estimating the cumulative distribution functions of the variables and by testing the equality of mean monthly and annual values before and after mid–1980s. In this analysis we considered 
[revised manuscript text omitted]